# Time-o1: Time-Series Forecasting Needs Transformed Label Alignment

**Hao Wang**[1][*]  **Licheng Pan**[1][*]  **Zhichao Chen**[2]  **Xu Chen**[3][†]
**Qingyang Dai**[4]  **Lei Wang**[3]  **Haoxuan Li**[5][†]  **Zhouchen Lin**[2,6,7][†]

[1]Xiaohongshu Inc.
[2]State Key Lab of General AI, School of Intelligence Science and Technology, Peking University
[3]Gaoling School of Artificial Intelligence, Renmin University of China
[4]Department of Control Science and Engineering, Zhejiang University
[5]Center for Data Science, Peking University
[6]Institute for Artificial Intelligence, Peking University
[7]Pazhou Laboratory (Huangpu), Guangzhou, Guangdong, China

## Abstract

Training time-series forecasting models poses unique challenges in loss function design. Most existing approaches adopt temporal mean squared error, but this study reveals two critical limitations: ❶ **it ignores the presence of label autocorrelation**, which biases it from the true label sequence likelihood; ❷ **it involves excessive number of tasks**, which complicates optimization, especially for long-term forecasting. To address these issues, we introduce Time-o1, a transform-enhanced lo**ss** function for **time**-series forecasting. The central idea is to transform the label sequence into decorrelated components with discriminated significance. Models are then trained to align the most significant components, thereby effectively mitigating label autocorrelation and reducing task amount. Experiments demonstrate that Time-o1 achieves state-of-the-art performance and is compatible with various forecast models. Code is available at https://github.com/Master-PLC/Time-o1.

## 1 Introduction

Time-series forecasting involves predicting future data from historical observations [62, 24] and has been applied across diverse domains, such as air quality prediction in meteorology [29], user behavior analysis in e-commerce [3], and process monitoring in manufacturing [49, 52]. To build effective forecasting models, two questions warrant investigation: *(1) How to design a neural network architecture to encode historical observations, and (2) How to devise a loss function to train the neural network.* Both are critical for forecast model performance.

Recent research has primarily focused on developing neural network architectures [57, 61]. The key challenge lies in exploiting the autocorrelation in history sequences. To this end, various architectures have been proposed [25, 41, 53, 31]. Current progress centers on a debate between Transformers and simple linear models. Transformers, equipped with self-attention mechanisms, offer superior scalability [28, 35, 30]. In contrast, linear models, which encapsulate temporal dynamics using linear layers, are straightforward to implement and demonstrate strong performance [67, 44, 65, 63]. These advancements showcase the rapid evolution in neural architecture design for time-series forecasting.

---

[*]This work was done in the internship at Xiaohongshu Inc. Both authors have equal contribution.
[†]Corresponding author.

39th Conference on Neural Information Processing Systems (NeurIPS 2025).

In contrast, the design of loss functions has received less attention [51, 40, 23]. Most studies adopt the temporal mean squared error (TMSE) as the loss function, which calculates the step-wise difference between the forecast and label sequences [28, 35]. While it is effective in various scenarios, it exhibits two limitations in time-series forecasting: ❶ **it ignores the presence of label autocorrelation**, treating each label step as independent, which renders it biased from the true likelihood of the label sequence [51]; ❷ **it involves excessive number of tasks**, where the task amount corresponds to the forecast horizon, which complicates optimization especially in long-term forecast scenarios [69].

To handle these challenges, we propose a transform-enhanced loss function for **time**-series forecasting (Time-o1). The key idea is to transform the label sequence into decorrelated components ranked by significance. By aligning the most significant decorrelated components, Time-o1 mitigates label autocorrelation and reduces the number of tasks. Our main contributions are summarized as follows:

- We formulate two critical challenges in designing loss functions for time-series forecasting: the label autocorrelation that induces bias, and the excessive number of tasks that impedes optimization.
- We propose Time-o1, a novel loss function for training time-series forecast models. It transforms label and forecast sequences into decorrelated components ranked by significance and subsequently align them, which effectively addresses challenges ❶-❷ with theoretical guarantees.
- We validate the efficacy of Time-o1 through extensive experiments, where Time-o1 consistently outperforms existing loss functions and enhances the performance of various forecast models.

## 2 Preliminaries

This paper focuses on the time-series forecasting problem [39, 56]. In general, we adhere to standard notational conventions: uppercase bold letters (*e.g.*, $\mathbf{X}$) denote matrices, lowercase bold letters (*e.g.*, $\mathbf{x}$) denote vectors, and lowercase normal letters (*e.g.*, $x$) denote scalars. Since the autocorrelation property central to our analysis manifests independently within each variate, we adopt the univariate setting for problem formulation [35], which generalizes naturally to the multivariate setting.

A time series, denoted by $\mathbf{s} = \{s_1, \ldots, s_M\} \in \mathbb{R}^M$, consists of a sequence of chronologically ordered observations. At any time step $n$, the history sequence is represented by $\mathbf{x} = [s_{n-H+1}, \ldots, s_n] \in \mathbb{R}^H$, and the corresponding label sequence is $\mathbf{y} = [s_{n+1}, \ldots, s_{n+T}] \in \mathbb{R}^T$, where H denotes the history length and T is the forecast horizon. The goal of time-series forecasting is to learn a model $g : \mathbb{R}^H \to \mathbb{R}^T$ that produces a forecast sequence $\hat{\mathbf{y}}$ closely matching the ground truth label sequence.

There are two aspects to building forecast models: (1) neural network architectures that effectively encode history sequences, and (2) loss functions for training these neural networks. While **this paper focuses on the loss function**, we provide a brief review of both aspects for contextualization.

### 2.1 Model architectures for time-series forecasting

Neural networks are widely employed to encode history sequences [32, 60] due to their ability to automatically model feature interactions and capture complex nonlinear autocorrelation [31, 12, 11]. Notable examples include recurrent neural networks (e.g., S4 [10], Mamba [9], P-sLSTM [16]), convolutional neural networks (e.g., SCINet [27], TimesNet [58], MICN [54]), and graph neural networks (e.g., MTGNN [34], StemGNN [2]), each tailored to encode the autocorrelations within input sequences. The current progress centers on comparisons between Transformer-based and linear architectures. Transformers (e.g., PatchTST [35], iTransformer [28], FreeFormer [66]) exhibit substantial scalability with increasing data size but entail high computational costs. In contrast, linear architectures (e.g., DLinear [67], RLinear [44], OLinear [65], TimeBase [13]) are generally more efficient but less scalable with larger datasets and struggle to handle varying input lengths.

### 2.2 Loss functions for time-series forecasting

Modern time-series models predominantly adopt the direct forecast paradigm, generating T-step forecasts simultaneously using a multi-output head [22, 28, 67]. The standard loss function is the temporal mean squared error (TMSE) between the forecast and label sequences, given by:

$$\mathcal{L}_{\mathrm{MSE}} = \left\| \mathbf{Y} - \hat{\mathbf{Y}} \right\|_2^2 = \sum_{n=1}^N \sum_{t=1}^T (y_{n,t} - \hat{y}_{n,t})^2, \tag{1}$$

where N is the number of samples. $\mathcal{L}_{\text{MSE}}$ is widespread in recent studies (e.g., FreTS [64], Fred-Former [37], iTransformer [28], DUET [41]); however, it proves biased in the presence of label autocorrelation [51]. To address this bias, one line of works performs shape alignment between the forecast and label sequences (e.g., Soft-DTW [4], Dilate [18], and STRIPE [19]). While these methods heuristically exploit label autocorrelation, they lack theoretical guarantees regarding unbiasedness and empirical evidence of improved performance [18]. A more recent and notable approach aligns the label and forecast sequence in the frequency domain [51, 23]; this strategy offers theoretical guarantees for bias reduction [51] and empirical improvements across scenarios [55, 20, 70].

## 3 Methodology

### 3.1 Motivation

The design of loss function is pivotal in training forecast models. Most previous studies employ $\mathcal{L}_{\text{MSE}}$ in (1) as the default loss function [28, 35, 65]. However, it suffers from two limitations stemming from the inherent properties of time series. ❶ **It ignores the presence of label autocorrelation**. Specifically, each observation in time-series is dependent on its past values [67]; this leads to label autocorrelation, i.e., different steps in the label sequence are correlated. However, $\mathcal{L}_{\text{MSE}}$ assumes each label step is independent, which disregards label autocorrelation and thus yields a biased loss function, as formulated in Theorem 3.1. ❷ **It involves excessive number of tasks.** Specifically, the number of forecast tasks in $\mathcal{L}_{\text{MSE}}$ corresponds directly to the forecast horizon T. While large forecast horizons are crucial for applications such as manufacturing (requiring long-horizon planning [50, 48]) and transportation (facilitating proactive traffic control [68, 33]), they introduce optimization challenges, e.g., gradient conflicts [69, 26], which impedes convergence and leads to suboptimal performance.

**Theorem 3.1** (Autocorrelation bias). *Given a univariate label sequence* $\mathbf{y} \in \mathbb{R}^{\text{T}}$ *where* $\boldsymbol{\Sigma} \in \mathbb{R}^{\text{T} \times \text{T}}$ *denotes the step-wise correlation coefficient, the loss function* $\mathcal{L}_{\text{MSE}}$ *in* (1) *is biased from the true negative log-likelihood of the label sequence, which is given by:*

$$\text{Bias} = \|\mathbf{y} - \hat{\mathbf{y}}\|_{\boldsymbol{\Sigma}^{-1}}^2 - \|\mathbf{y} - \hat{\mathbf{y}}\|^2 - \frac{1}{2}\log|\boldsymbol{\Sigma}|. \tag{2}$$

*where* $\|\mathbf{v}\|_{\boldsymbol{\Sigma}^{-1}}^2 = \mathbf{v}^\top \boldsymbol{\Sigma}^{-1} \mathbf{v}$. *The bias vanishes if different steps in* $\mathbf{y}$ *are decorrelated.*[3]

Designing a loss function to handle the two limitations is challenging. Our previous work [51] proposes a frequency-domain loss, which transforms the label and forecast sequences into frequency components and then aligns them. It is motivated by Theorem 3.1: bias vanishes if different components are decorrelated. However, the decorrelation of frequency components holds only when the forecast horizon $\text{T} \to \infty$ (see Theorem 3.3 in [51]). In real-world settings with a finite horizon, frequency components remain correlated, rendering FreDF ineffective to fully eliminate bias. Moreover, optimization difficulty remains, since transforming to the frequency domain retains the label length. *Consequently, FreDF does not fully address the limitations ❶-❷ discussed in this paper.*

Given the significance of loss function in training forecast models and the limitations of existing methods, it is compelling to develop an innovative loss function to address the limitations and advance forecast performance. Concretely, two questions warrant investigation. *How to devise a loss function to eliminate autocorrelation bias and reduce task amount? Does it improve forecast performance?*

### 3.2 Transforming label sequence with optimized projection matrix

In this section, we present a method for transforming label sequences into latent components to eliminate autocorrelation and distinguish significant components. Suppose $\mathbf{Y} \in \mathbb{R}^{\text{N} \times \text{T}}$ contains univariate label sequences of N samples, $\mathbf{P} = [\mathbf{p}_1, \mathbf{p}_2, ..., \mathbf{p}_{\text{K}}]$ is the projection matrix; the components are produced as $\mathbf{Z} = \mathbf{YP}$. The target is for $\mathbf{Z}$ to be decorrelated and ranked by significance. For example, FreDF specifies $\mathbf{P}$ as a Fourier matrix, which does not adapt to specific data properties and thus fails to decorrelate the components and distinguish the significant components[4].

---

[3]The pioneering work [51] identifies the bias under the first-order Markov assumption on the label sequence. This study generalizes this bias without the first-order Markov assumption.

[4]In the multivariate case, different variates can be treated separately to produce decorrelated components.

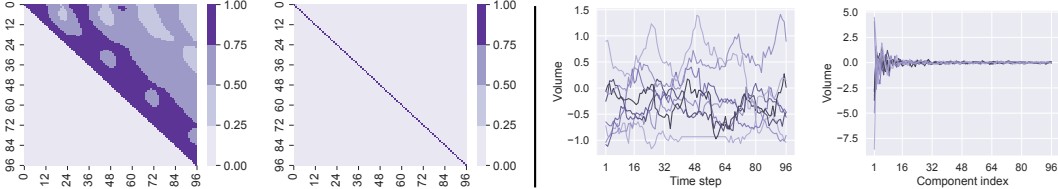

(a) Autocorrelation in label sequence and components. (b) Volume of label sequences and latent components.

Figure 1: The autocorrelations and volumes in the label sequence $\mathbf{Y}$ and latent components $\mathbf{Z}$.

A natural approach to obtaining $\mathbf{P}$ with the desired properties is constructing optimization problem with constraints. Specifically, to find the $k$-th component, the projection vector $\mathbf{p}_k^*$ can be defined as:

$$\mathbf{p}_k^* = \underset{\mathbf{p}_k}{\arg\max} \quad (\mathbf{Y}\mathbf{p}_k)^\top (\mathbf{Y}\mathbf{p}_k)$$
$$\text{subject to} \quad \begin{cases} \|\mathbf{p}_k\|^2 = 1 \\ \mathbf{p}_k^\top \mathbf{p}_{k'} = 0, \ \forall k' < k \quad \text{if } k > 1 \end{cases} \tag{3}$$

where $\mathbf{z}_k = \mathbf{Y}\mathbf{p}_k$ is the $k$-th component, the normalization constraint $\|\mathbf{p}_k\|^2 = 1$ is imposed to avoid trivial solution: $\mathbf{p}_k \to \infty$. The optimization target is to maximize the variance of $\mathbf{z}_k$, *which is equivalent to maximizing its significance, as components with larger variance contain richer information*. For $k > 1$, the projection axis is required to be orthogonal to the previous axes to avoid redundancy. By solving the optimizations above from $k = 1$ to $\mathrm{K} \le \mathrm{T}$ sequentially, we obtain the projection matrix $\mathbf{P}^* = [\mathbf{p}_1^*, ..., \mathbf{p}_K^*]$. The components are then produced as $\mathbf{Z} = \mathbf{Y}\mathbf{P}^*$.

**Lemma 3.2** (Decorrelated components). *Suppose $\mathbf{Y} \in \mathbb{R}^{\mathrm{N} \times \mathrm{T}}$ contains normalized label sequences for $\mathrm{N}$ samples, $\mathbf{Z} = [\mathbf{z}_1, ..., \mathbf{z}_K]$ are the obtained components; for any $k \ne k'$, we have $\mathbf{z}_k^\top \mathbf{z}_{k'} = 0$.*

**Lemma 3.3.** *The projection matrix $\mathbf{P}^*$ can be obtained via singular value decomposition (SVD): $\mathbf{Y} = \mathbf{U}\boldsymbol{\Lambda}(\mathbf{P}^*)^\top$, where $\mathbf{U} \in \mathbb{R}^{\mathrm{N} \times \mathrm{N}}$ and $\mathbf{P}^* \in \mathbb{R}^{\mathrm{K} \times \mathrm{K}}$ consist of singular vectors, and the diagonal of $\boldsymbol{\Lambda} \in \mathbb{R}^{\mathrm{N} \times \mathrm{K}}$ consists of singular values.*

**Theoretical Justification.** By Theorem 3.1, the autocorrelation bias vanishes as the label correlations are eliminated. Since the obtained components $\mathbf{Z}$ are decorrelated (Lemma 3.2), applying step-wise difference to align them suffers from little autocorrelation bias. Moreover, component significance decreases from $\mathbf{z}_1$ to $\mathbf{z}_K$ as they are derived under sequentially added constraints. Furthermore, $\mathbf{P}^*$ can be computed via SVD (Lemma 3.3), offering an efficient strategy to obtain $\mathbf{P}^*$.

**Case study.** To showcase the implications of the obtained components, a case study was conducted on the ETTh1 dataset. Implementation details are provided in Appendix A. The results are illustrated in Fig. 1, with key observations summarized as follows:

- **Decorrelation property:** Fig. 1 (a) compares the autocorrelation volume in the label sequence and the transformed components obtained by (3). In the left panel, the value at row $i$ and column $j$ represents the correlation between the $i$-th and $j$-th steps in the label sequence. A large number of non-diagonal elements exhibit substantial values, with approximately 50.5% exceeding 0.25, indicating significant label autocorrelation. In contrast, the right panel shows negligible values for the non-diagonal elements. This demonstrates that transforming the label sequence into components effectively eliminates correlation, which empirically validates Lemma 3.2.

- **Significance discrimination:** Fig. 1 (b) compares the variance of the label sequence and the transformed components in (3). In the left panel, the variance of the label sequence is almost uniform across different steps (ranging from -1.5 to 1.5), indicating that different steps contribute similarly to the overall information and are almost equally significant. In the right panel, a small number of components has large variance. This demonstrates that the transform yields components with ranked significance. As a result, one can balance minor information loss with substantial decreases in optimization complexity by concentrating learning on the most significant components.

The transformation is highly inspired by principal component analysis (PCA) [36]. However, one key distinction warrants emphasis. Existing works dominantly employ principal component analysis

on *input features* for obtaining informative representations [8, 6], in contrast, we adapt it to *label sequence*, specifically aiming to reduce autocorrelation bias and simplify optimization for time-series forecasting. To our knowledge, this remains a technically innovative strategy.

### 3.3 Model implementation

In this section, we present the implementation details of Time-o1. The approach centers on extracting the latent components from the label sequence, then optimizing the forecast model using the most significant components.

Given N history sequences $\mathbf{X} \in \mathbb{R}^{N \times H}$ and label sequences $\mathbf{Y} \in \mathbb{R}^{N \times T}$, the forecast model generates forecast sequences $\hat{\mathbf{Y}} = g(\mathbf{X})$. In line with prior practices [28, 67, 37], we standardize $\mathbf{Y}$ (step 1), which is a prerequisite for ensuring the decorrelation of the resulting components (see Lemma 3.2). Next, following Lemma 3.3, we compute the optimal projection matrix $\mathbf{P}^*$ by applying SVD to $\mathbf{Y}$, retaining only the K right singular vectors corresponding to the largest singular values (steps 2-3). Subsequently, both $\mathbf{Y}$ and $\hat{\mathbf{Y}}$ are projected into the latent component space (step 4). In this space, significance is strictly ordered: the first column captures the greatest significance (variance), which successively diminishes across subsequent columns.

---

**Algorithm 1** The workflow of Time-o1.

---

**Input**: $\hat{\mathbf{Y}}$: forecast sequences, $\mathbf{Y}$: label sequences.
**Parameter**: $\alpha$: the relative weight of the transformed loss, $\gamma$: the ratio of retained components.
**Output**: $\mathcal{L}_{\alpha,\gamma}$: the obtained loss function.

1: $\mathbf{Y} \leftarrow \text{standardize}(\mathbf{Y})$.
2: $\text{K} \leftarrow \text{round}(\gamma \cdot \text{T})$
3: $\mathbf{P}^* \leftarrow \text{SVD}(\mathbf{Y}; \text{K})$
4: $\mathbf{Z} \leftarrow \mathbf{Y}\mathbf{P}^*, \hat{\mathbf{Z}} \leftarrow \hat{\mathbf{Y}}\mathbf{P}^*$
5: $\mathcal{L}_{\text{ortho},\gamma} \leftarrow \|\hat{\mathbf{Z}} - \mathbf{Z}\|_1$
6: $\mathcal{L}_{\text{MSE}} \leftarrow \|\hat{\mathbf{Y}} - \mathbf{Y}\|_2^2$
7: $\mathcal{L}_{\alpha,\gamma} := \alpha \cdot \mathcal{L}_{\text{ortho},\gamma} + (1-\alpha) \cdot \mathcal{L}_{\text{MSE}}$.

---

Afterwards, we compute the transformed loss to align $\mathbf{Y}$ and $\hat{\mathbf{Y}}$ in the transformed space (step 5):

$$\mathcal{L}_{\text{ortho},\gamma} := \|\mathbf{Z} - \mathbf{Z}\|_1, \tag{4}$$

where $\gamma$ controls the ratio of components retained, such that $\text{K} = \text{round}(\gamma \cdot \text{T})$; $\|\cdot\|_1$ computes the sum of element-wise absolute differences. Notably, we use the $\ell_1$ norm instead of the squared norm following [51], since different latent components vary greatly in scale (see Fig. 1), which makes the squared norm unstable. The $\ell_1$ norm provides a more stable and robust optimization landscape.

Finally, we fuse the two loss functions, with $0 \le \alpha \le 1$ controlling the relative contribution (step 7):

$$\mathcal{L}_{\alpha,\gamma} := \alpha \cdot \mathcal{L}_{\text{ortho},\gamma} + (1-\alpha) \cdot \mathcal{L}_{\text{MSE}}. \tag{5}$$

By transforming both forecasts and labels into decorrelated components, Time-o1 effectively reduces autocorrelation bias. By focusing exclusively on the most significant components, Time-o1 reduces optimization difficulty with minimal information loss. Time-o1 is model-agnostic, offering practitioners the flexibility to employ the most suitable forecast model for each specific scenario.

## 4 Experiments

To demonstrate the efficacy of Time-o1, there are six aspects empirically investigated:

1. **Performance:** *Does Time-o1 work?* We compare Time-o1 with state-of-the-art baselines using public datasets on long-term forecasting in Section 4.2 and short-term forecasting tasks in Appendix E.1. Moreover, we compare Time-o1 with other loss functions in Section 4.3.

2. **Gain:** *How does it work?* Section 4.4 offers an ablative study to dissect the contributions of the individual factors of Time-o1, elucidating their roles in enhancing forecast accuracy.

3. **Generality:** *Does it support other forecast models?* Section 4.5 verifies the adaptability of Time-o1 across different forecast models, with additional results in Appendix E.4.

4. **Flexibility:** *Does it support alternative transformations?* Section 4.5 also investigates generating latent components with other transformations to showcase flexibility of implementation.

5. **Sensitivity:** *Does it require careful fine-tuning?* Section 4.6 presents a sensitivity analysis of the hyperparameter $\alpha$, where Time-o1 maintains efficacy across a broad range of parameter values.

6. **Efficiency:** *Is it efficient?* Section D shows the running time of Time-o1 in diverse settings.

Table 1: Long-term forecasting performance.

| Models | Time-o1 (Ours) | | Fredformer (2024) | | iTransformer (2024) | | FreTS (2023) | | TimesNet (2023) | | MICN (2023) | | TiDE (2023) | | DLinear (2023) | | FEDformer (2022) | | Autoformer (2021) | | Transformer (2017) | |
|---|---|---|---|---|---|---|---|---|---|---|---|---|---|---|---|---|---|---|---|---|---|---|
| Metrics | MSE | MAE | MSE | MAE | MSE | MAE | MSE | MAE | MSE | MAE | MSE | MAE | MSE | MAE | MSE | MAE | MSE | MAE | MSE | MAE | MSE | MAE |
| ETTm1 | **0.380** | **0.393** | 0.387 | 0.398 | 0.411 | 0.414 | 0.414 | 0.421 | 0.438 | 0.430 | 0.396 | 0.421 | 0.413 | 0.407 | 0.403 | 0.407 | 0.442 | 0.457 | 0.526 | 0.491 | 0.799 | 0.648 |
| ETTm2 | **0.272** | **0.317** | 0.280 | 0.324 | 0.295 | 0.336 | 0.316 | 0.365 | 0.302 | 0.334 | 0.308 | 0.364 | 0.286 | 0.328 | 0.342 | 0.392 | 0.308 | 0.354 | 0.315 | 0.358 | 1.662 | 0.917 |
| ETTh1 | **0.431** | **0.429** | 0.447 | 0.434 | 0.452 | 0.448 | 0.489 | 0.474 | 0.472 | 0.463 | 0.533 | 0.519 | 0.448 | 0.435 | 0.456 | 0.453 | 0.447 | 0.470 | 0.477 | 0.483 | 0.983 | 0.774 |
| ETTh2 | **0.359** | **0.388** | 0.377 | 0.402 | 0.386 | 0.407 | 0.524 | 0.496 | 0.409 | 0.420 | 0.620 | 0.546 | 0.378 | 0.401 | 0.529 | 0.499 | 0.452 | 0.461 | 0.448 | 0.460 | 2.688 | 1.291 |
| ECL | **0.170** | **0.260** | 0.191 | 0.284 | 0.179 | 0.270 | 0.199 | 0.288 | 0.212 | 0.306 | 0.192 | 0.302 | 0.215 | 0.292 | 0.212 | 0.301 | 0.214 | 0.328 | 0.249 | 0.354 | 0.265 | 0.358 |
| Traffic | **0.419** | **0.280** | 0.486 | 0.336 | 0.426 | 0.285 | 0.538 | 0.330 | 0.631 | 0.338 | 0.529 | 0.312 | 0.624 | 0.373 | 0.625 | 0.384 | 0.640 | 0.398 | 0.662 | 0.416 | 0.692 | 0.379 |
| Weather | **0.241** | **0.280** | 0.261 | 0.282 | 0.269 | 0.289 | 0.249 | 0.293 | 0.271 | 0.295 | 0.264 | 0.321 | 0.272 | 0.291 | 0.265 | 0.317 | 0.326 | 0.372 | 0.319 | 0.365 | 0.699 | 0.601 |
| PEMS03 | **0.097** | **0.208** | 0.146 | 0.260 | 0.122 | 0.233 | 0.149 | 0.261 | 0.126 | 0.230 | 0.106 | 0.223 | 0.316 | 0.370 | 0.216 | 0.322 | 0.152 | 0.275 | 0.411 | 0.475 | 0.122 | 0.226 |
| PEMS08 | **0.141** | **0.237** | 0.171 | 0.271 | 0.149 | 0.247 | 0.174 | 0.275 | 0.152 | 0.243 | 0.153 | 0.258 | 0.318 | 0.378 | 0.249 | 0.332 | 0.226 | 0.312 | 0.422 | 0.456 | 0.240 | 0.261 |

*Note*: We fix the input length as 96 following [28]. **Bold** and underlined denote best and second-best results, respectively. *Avg* indicates average results over forecast horizons: T=96, 192, 336 and 720. Time-o1 employs the top-performing baseline on each dataset as its underlying forecast model.

## 4.1 Setup

**Datasets.** In this work, we conduct experiments on ETT (4 subsets), ECL, Traffic, Weather, and PEMS [28] for long-term forecasting task, and M4 for short-term forecasting task [58]. All datasets are split chronologically into training, validation, and testing sets following their official settings.

**Baselines.** We compare Time-o1 with a range of established models, including Transformer [45], Autoformer [59], FEDformer [71], iTransformer [28], Fredformer [37], DLinear [67], TiDE [5], FreTS [64], TimesNet [58], and MICN [54]. As a loss function, Time-o1 is model-agnostic and can be integrated with any model architecture. By default, Time-o1 employs the best-performing baseline model on each dataset as its underlying model architecture for fair comparison.

**Implementation.** The baseline models are reproduced using the scripts provided by Fredformer [37]. All baseline models are trained using the Adam [14] optimizer to minimize $\mathcal{L}_{\mathrm{MSE}}$ in (1). Following the prestigious benchmark [38], the dropping-last trick is disabled during the test phase. When integrating Time-o1 to enhance an established model, we adhere to the associated hyperparameter settings in the public benchmark [37, 28], only tuning $\alpha$, $\gamma$ and learning rate conservatively. Experiments are conducted on Intel(R) Xeon(R) Platinum 8383C CPUs and NVIDIA RTX H100 GPUs.

## 4.2 Overall performance

Table 1 presents the long-term forecasting results. Time-o1 consistently improves base model performance. For example, on ETTh1, it reduces Fredformer's MSE by 0.016. Similar gains across other datasets further validate its effectiveness. These results suggest that modifying the loss function can yield improvements comparable to, or even exceeding, those from architectural advancements. We attribute this to two key aspects of Time-o1: its decorrelation property for eliminating autocorrelation bias and its discrimination on significant components for simplifying optimization.

**Showcases.** We visualize the forecast sequences and the generated components to showcase the improvements of Time-o1 in forecast quality. A snapshot on ETTm2 with historical window H = 96 and forecast horizon T = 336 is depicted in Fig. 2. Although the model trained using canonical DF captures general trends, its forecast struggles with large variations (e.g., peaks within steps 100-400). This reflects its difficulty in modeling significant, high-variance components. In contrast, Time-o1, by explicitly discriminating and aligning these significant components, generates a forecast that accurately captures these large variations, including the peaks within steps 100-400.

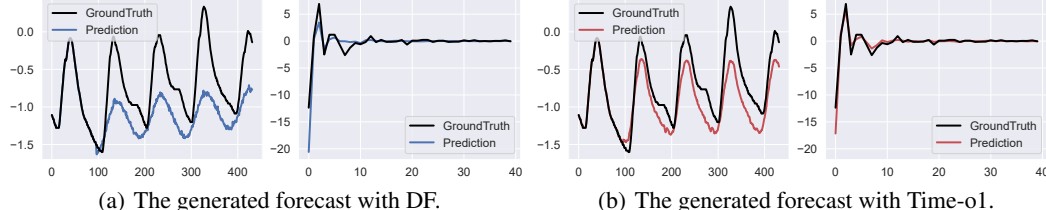

(a) The generated forecast with DF.  (b) The generated forecast with Time-o1.

Figure 2: The visualization of label and forecast sequences generated by models trained with TMSE versus Time-o1. In both (a) and (b), the left panels display the time-domain sequences ($\mathbf{Y}$ and $\hat{\mathbf{Y}}$), while the right panels illustrate their corresponding latent components ($\mathbf{Z}$ and $\hat{\mathbf{Z}}$).

Table 2: Comparable results with other loss functions for time-series forecast.

| Loss | | **Time-o1** | | FreDF | | Koopman | | Dilate | | Soft-DTW | | DPTA | | DF | |
|---|---|---|---|---|---|---|---|---|---|---|---|---|---|---|---|
| Metrics | | MSE | MAE | MSE | MAE | MSE | MAE | MSE | MAE | MSE | MAE | MSE | MAE | MSE | MAE |
| Fredformer | ETTm1 | **0.379** | **0.393** | 0.384 | 0.394 | 0.389 | 0.400 | 0.389 | 0.400 | 0.397 | 0.402 | 0.396 | 0.402 | 0.387 | 0.398 |
| | ETTh1 | **0.431** | **0.429** | 0.438 | 0.434 | 0.452 | 0.443 | 0.453 | 0.442 | 0.460 | 0.449 | 0.460 | 0.449 | 0.447 | 0.434 |
| | ECL | **0.178** | **0.270** | 0.179 | 0.272 | 0.190 | 0.282 | 0.187 | 0.280 | 0.206 | 0.298 | 0.202 | 0.294 | 0.191 | 0.284 |
| | Weather | **0.255** | **0.276** | 0.256 | 0.277 | 0.257 | 0.279 | 0.258 | 0.280 | 0.261 | 0.280 | 0.260 | 0.280 | 0.261 | 0.282 |
| iTransformer | ETTm1 | **0.395** | **0.401** | 0.405 | 0.405 | 0.413 | 0.416 | 0.407 | 0.412 | 0.417 | 0.415 | 0.416 | 0.415 | 0.411 | 0.414 |
| | ETTh1 | **0.438** | **0.434** | 0.442 | 0.437 | 0.455 | 0.451 | 0.452 | 0.448 | 0.470 | 0.457 | 0.463 | 0.454 | 0.452 | 0.448 |
| | ECL | **0.170** | **0.260** | 0.176 | 0.264 | 0.178 | 0.269 | 0.178 | 0.269 | 0.175 | 0.266 | 0.177 | 0.267 | 0.179 | 0.270 |
| | Weather | **0.251** | **0.272** | 0.257 | 0.276 | 0.289 | 0.313 | 0.286 | 0.309 | 0.292 | 0.316 | 0.291 | 0.313 | 0.269 | 0.289 |

*Note*: **Bold** and underlined denote best and second-best results, respectively. The reported results are averaged over forecast horizons: T=96, 192, 336 and 720.

## 4.3 Loss function comparison

Table 2 compares Time-o1 against other time-series loss functions: FreDF [51], Koopman [17], Dilate [18], Soft-DTW [4], and DPTA [42]. We integrate their official implementations into Fredformer and iTransformer. Overall, shape alignment losses (Dilate, Soft-DTW, DPTA) offer little performance gain over canonical DF (using TMSE loss), consistent with the findings in [18]. This phenomenon is rationalized by the fact that they do not mitigate the label autocorrelation nor reduce task amounts for simplifying optimization. FreDF improves performance by partly addressing autocorrelation bias. However, as discussed in Section 3.1, FreDF does not fully eliminate this bias, nor does it distinguish significant components to simplify the optimization landscape. Time-o1 directly addresses these two limitations of FreDF, leading to its superior overall performance.

## 4.4 Ablation studies

Table 3 presents an ablation study dissecting the contributions of critical factors in Time-o1: the decorrelation property and the task reduction effect. The main findings are summarized as follows.

- Time-o1$^{\dagger}$ improves DF by reducing the number of tasks to optimize. To this end, it employs a randomized matrix as the projection matrix to generate components and aligns only a subset of the obtained components. The involution ratio $\gamma$ is finetuned on the validation set. It consistently improves over DF (e.g., $-0.012$ MAE on Weather). This demonstrates that reducing tasks with a minimal loss of label information can reduce optimization difficulty and improve performance.

- Time-o1$^{\ddagger}$ improves DF by aligning decorrelated components. To this end, the loss function is calculated in (5) with $\gamma = 1$. It also outperforms DF, achieving the second-best results overall. This demonstrates aligning decorrelated label components to mitigate bias benefits forecast performance.

- Time-o1 integrates both factors above by aligning the most significant decorrelated components and achieves the best overall performance, demonstrating the synergistic effect of these two factors.

Table 3: Ablation study results.

| Model | Decorrelation | Reduction | Data | T=96 | | T=192 | | T=336 | | T=720 | | Avg | |
|---|---|---|---|---|---|---|---|---|---|---|---|---|---|
| | | | | MSE | MAE | MSE | MAE | MSE | MAE | MSE | MAE | MSE | MAE |
| DF | ✗ | ✗ | ETTm1 | 0.326 | 0.361 | 0.365 | 0.382 | 0.396 | 0.404 | 0.459 | 0.444 | 0.387 | 0.398 |
| | | | ETTh1 | 0.377 | 0.396 | 0.437 | 0.425 | 0.486 | 0.449 | 0.488 | 0.467 | 0.447 | 0.434 |
| | | | ECL | 0.150 | 0.242 | 0.168 | 0.259 | 0.182 | 0.274 | 0.214 | 0.304 | 0.179 | 0.270 |
| | | | Weather | 0.174 | 0.228 | 0.213 | 0.266 | 0.270 | 0.316 | 0.337 | 0.362 | 0.249 | 0.293 |
| Time-o1† | ✗ | ✓ | ETTm1 | 0.338 | 0.366 | 0.369 | 0.383 | 0.397 | 0.403 | 0.458 | 0.441 | 0.391 | 0.398 |
| | | | ETTh1 | 0.376 | 0.395 | 0.437 | 0.430 | 0.478 | 0.450 | _0.469_ | 0.467 | 0.440 | 0.436 |
| | | | ECL | 0.150 | 0.239 | 0.164 | 0.253 | 0.178 | 0.268 | 0.210 | 0.296 | 0.175 | 0.264 |
| | | | Weather | _0.170_ | **0.216** | 0.213 | 0.259 | 0.262 | _0.300_ | 0.332 | _0.351_ | 0.244 | _0.281_ |
| Time-o1‡ | ✓ | ✗ | ETTm1 | _0.324_ | _0.359_ | _0.362_ | _0.379_ | _0.390_ | _0.400_ | _0.451_ | _0.438_ | _0.382_ | _0.394_ |
| | | | ETTh1 | _0.373_ | _0.395_ | _0.433_ | _0.423_ | _0.476_ | _0.445_ | 0.474 | _0.463_ | 0.439 | _0.431_ |
| | | | ECL | _0.147_ | _0.238_ | _0.162_ | _0.252_ | _0.174_ | _0.267_ | _0.205_ | _0.294_ | _0.172_ | _0.263_ |
| | | | Weather | 0.172 | 0.220 | _0.211_ | _0.259_ | _0.261_ | 0.301 | _0.331_ | 0.353 | _0.244_ | 0.283 |
| Time-o1 | ✓ | ✓ | ETTm1 | **0.321** | **0.357** | **0.360** | **0.378** | **0.389** | **0.400** | **0.447** | **0.435** | **0.379** | **0.393** |
| | | | ETTh1 | **0.368** | **0.391** | **0.424** | **0.422** | **0.467** | **0.441** | **0.465** | **0.463** | **0.431** | **0.429** |
| | | | ECL | **0.145** | **0.235** | **0.159** | **0.249** | **0.173** | **0.264** | **0.203** | **0.292** | **0.170** | **0.260** |
| | | | Weather | **0.169** | _0.219_ | **0.210** | **0.258** | **0.259** | **0.297** | **0.327** | **0.349** | **0.241** | **0.280** |

*Note*: **Bold** and underlined denote best and second-best results, respectively.

Table 4: Varying transformations results.

| | ECL | | | | Weather | | | |
|---|---|---|---|---|---|---|---|---|
| Transformation | MSE | Δ | MAE | Δ | MSE | Δ | MAE | Δ |
| None | 0.179 | - | 0.270 | - | 0.249 | - | 0.293 | - |
| RPCA | _0.171_ | 4.31% ↓ | _0.261_ | 3.16% ↓ | _0.244_ | 1.78% ↓ | _0.286_ | 2.38% ↓ |
| SVD | 0.175 | 2.24% ↓ | 0.264 | 2.18% ↓ | 0.248 | 0.34% ↓ | 0.290 | 0.93% ↓ |
| FA | 0.175 | 2.35% ↓ | 0.265 | 1.82% ↓ | 0.245 | 1.35% ↓ | 0.287 | 1.97% ↓ |
| Ours | **0.170** | **4.86%↓** | **0.260** | **3.57%↓** | **0.241** | **2.94%↓** | **0.280** | **4.28%↓** |

*Note*: Δ refers to the relative error reduction compared to the baseline (None). **Bold** and underlined denote best and second-best results.

## 4.5 Generalization studies

In this section, we investigate the utility of Time-o1 with different transformation strategies and forecast models, to showcase the generality of Time-o1. In the bar-plots, the forecast errors are averaged over forecast lengths (96, 192, 336, 720), with error bars as 50% confidence intervals.

**Varying transformations.** We select alternative approaches to transform the label sequence into latent components and report the forecast performance in Table 4. The selected transformation methods include robust principal component analysis (RPCA) [1], SVD [7], and factor analysis [15]. Noting that the output of SVD yields components here, not a projection matrix as in Section 3.2. Implementation details are in Appendix C. Overall, all these transformation methods outperform canonical DF without transformation. However, the components obtained by these methods, including RPCA, cannot be guaranteed to be decorrelated. Consequently, autocorrelation bias may persist. In contrast, our approach ensures full decorrelation of the derived components (see Lemma 3.2), effectively addressing autocorrelation bias and leading to the best overall performance.

**Varying forecast models.** We explore the versatility of Time-o1 in augmenting representative forecast models: Fredformer [37], iTransformer [28], FreTS [64], and DLinear [67]. As illustrated in Fig. 3, Time-o1 improves forecast performance in all cases. For instance, on the Weather dataset, iTransformer and FreTS with Time-o1 achieve substantial reductions in MSE—up to 6.9% and 2.9%, respectively. Further evidence of Time-o1's versatility can be found in Appendix E.4. These results confirm Time-o1's potential as a plug-and-play strategy to enhance diverse forecast models.

## 4.6 Hyperparameter sensitivity

In this section, we examine the impact of critical hyperparameters on the performance of Time-o1. The results are presented in Table 5 and Table 6. Additional trends across different datasets and forecast lengths are provided in Appendix E.5. The primary observations are summarized as follows:

- The coefficient $\alpha$ determines the relative importance of the proposed transform-enhanced loss in (5). When $\alpha$ is set to 1, Time-o1 exclusively uses the transform-enhanced loss. Overall, increasing $\alpha$ from 0 to 1 leads to improved forecast accuracy, with the best results typically achieved when

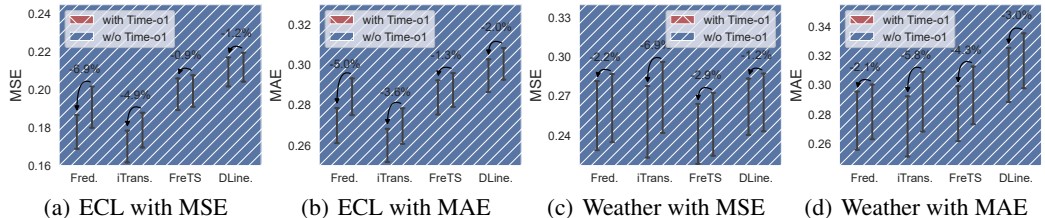

|   (a) ECL with MSE   |   (b) ECL with MAE   |   (c) Weather with MSE   |   (d) Weather with MAE   |

Figure 3: Improvement of Time-o1 applied to different forecast models, shown with colored bars for means over forecast lengths (96, 192, 336, 720) and error bars for 50% confidence intervals.



Table 5: Hyperparameter results on $\alpha$.

|          | ETTm1  |        | ETTh2  |        | Weather |        |
|----------|--------|--------|--------|--------|---------|--------|
| $\alpha$ | MSE    | MAE    | MSE    | MAE    | MSE     | MAE    |
| 0        | 0.3867 | 0.3979 | 0.3766 | 0.4019 | 0.2486  | 0.2930 |
| 0.3      | 0.3871 | 0.3983 | 0.3742 | 0.3982 | 0.2439  | 0.2851 |
| 0.5      | 0.3864 | 0.3976 | 0.3703 | 0.3964 | **0.2432** | **0.2833** |
| 0.7      | **0.3831** | 0.3959 | 0.3674 | **0.3943** | 0.2433 | 0.2849 |
| 1        | 0.3850 | **0.3933** | **0.3606** | 0.3890 | 0.2753 | 0.3209 |

*Note*: **Bold** and underlined denote best and second-best results.

Table 6: Hyperparameter results on $\gamma$.

|          | ETTm1  |        | ETTh2  |        | Weather |        |
|----------|--------|--------|--------|--------|---------|--------|
| $\gamma$ | MSE    | MAE    | MSE    | MAE    | MSE     | MAE    |
| 0.1      | 0.3915 | 0.4002 | 0.3816 | 0.4029 | 0.2437  | 0.2845 |
| 0.3      | 0.3849 | 0.3964 | 0.3694 | 0.3961 | **0.2424** | **0.2825** |
| 0.5      | 0.3817 | 0.3943 | 0.3651 | 0.3923 | 0.2466  | 0.2877 |
| 0.7      | **0.3798** | **0.3930** | **0.3603** | **0.3886** | 0.2443 | 0.2861 |
| 1        | 0.3814 | 0.3940 | 0.3624 | 0.3903 | 0.2491 | 0.2924 |

*Note*: **Bold** and underlined denote best and second-best results.



$\alpha$ is close to 1. The performance improvement is significant, e.g., MSE reduction on ETTh2 by 0.016, showcasing the utility of the transform-enhanced loss to improve forecast performance.

- The coefficient $\gamma$ determines the ratio of retained components for training. The results demonstrate that preserving all label information ($\gamma = 1$) does not necessarily yield optimal performance. Instead, the best results are obtained at $\gamma < 1$, such as 0.7 on ETTs and 0.3 on Weather. This phenomenon can be attributed to the trade-off between information loss and optimization complexity: by focusing on the top components, optimization is simplified due to reduced task amount. Besides, since the top components contain most information, the information loss by dropping the other components is negligible. These factors synergistically contribute to improved performance.

## 5 Conclusion

In this study, we highlight the importance of designing loss functions for training time-series forecasting models. Two critical challenges are formulated: label autocorrelation, which induces bias, and number of tasks, which determines optimization complexity. To address these challenges, we introduce a model-agnostic loss function called Time-o1. This method transforms the label sequence into decorrelated components with discernible significance. Then, it trains forecast models to align the most significant components, which simultaneously mitigates label autocorrelation and reduces task amount. Empirically, Time-o1 improves forecast performance across diverse datasets.

***Limitations & future works.*** In this work, we investigate the challenges of label autocorrelation and excessive number of tasks in time-series forecasting. Nevertheless, these issues also manifest in areas such as speech generation, target recognition, and dense image prediction. Applying Time-o1 in these contexts is a promising avenue for future research. Additionally, history sequence also exhibits autocorrelation and contains redundancy. Transforming inputs to derive decorrelated, compact representations could offer additional performance gains and also warrants investigation.

## Acknowledgments

Z. Lin was supported by the NSF China (No. 62276004) and the State Key Laboratory of General Artificial Intelligence. H. Li was supported by National Natural Science Foundation of China (623B002).

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

# A   On the Implementation Details of Label Correlation Estimation

In this section, we introduce the motivation and implementation details of the label autocorrelation estimation techniques in Fig. 1. Measuring label autocorrelation $y_t \rightarrow y_{t'}$ is indeed challenging due to the presence of confounding effect [47, 46, 21]. Specifically, the fork structure $y_t \leftarrow \mathbf{x} \rightarrow y_{t'}$ introduces spurious correlations between $y_t$ and $y_{t'}$, thereby distorting the true strength of the label autocorrelation $y_t \rightarrow y_{t'}$ of interest. This structural confounding undermines the validity of traditional measures such as Pearson correlation for quantifying label autocorrelation.

The previous work [51] employs the double machine learning (DML) method to estimate the ground-truth correlation while mitigating the influence of the fork structure. We adopt this in our experiments. DML is a statistical technique designed to estimate the causal effect of a treatment on an outcome while controlling for fork variables. Specifically, suppose we have a treatment variable $\mathcal{T}$, an outcome variable $\mathcal{Y}$, and a set of fork variables $\mathcal{X}$. The goal is to estimate the causal effect of $\mathcal{T}$ on $\mathcal{Y}$ while controlling for the influence of $\mathcal{X}$. To this end, DML first orthogonalizes both the treatment and outcome with respect to the fork variables. Two parametric models are employed to predict the treatment and outcome based on the fork variables. These predictions capture the impact of $\mathcal{X}$ on $\mathcal{Y}$ and $\mathcal{T}$. Subsequently, such impact of $\mathcal{X}$ is eliminated by calculating the residuals. Finally, the DML method regresses the outcome residuals on the treatment residuals, thereby measuring the causal effect of $\mathcal{T}$ on $\mathcal{Y}$ while removing the influence of the fork variables.

In our experiments, we measure label autocorrelation by treating the input sequence $\mathbf{x}$ as the fork variable and different steps of the label sequences $y_t$ and $y_{t'}$ as the treatment and outcome variables, respectively. Then, we estimate the treatment effect of $y_t$ on $y_{t'}$ controlling $\mathbf{x}$. Similarly, when measuring the correlation between different components, we use different components $z_k$ and $z_{k'}$ as the treatment and outcome variables. Linear regression model is employed as the parametric model for both the treatment and outcome variables for efficiency, which is consistent with [51].

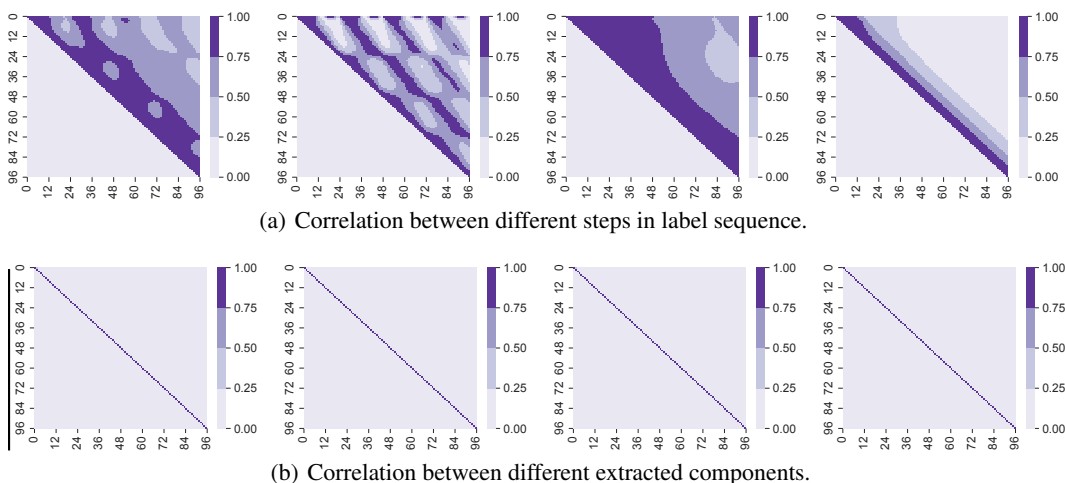

(a) Correlation between different steps in label sequence.

(b) Correlation between different extracted components.

Figure 4: The label autocorrelation in the original label sequence and the extracted components. The datasets are ETTh1, ETTh2, ETTm1, and Weather from left to right. The forecast length is set to 96.

To further complement the case study in Fig. 1, we analyzed the correlation matrices of the label sequences and the extracted components across multiple datasets, with the results presented in Fig. 4. The main observations are summarized as follows.

- Panel (a) displays the correlation matrix of the label sequence, characterized by substantial non-diagonal elements, which highlight the strong autocorrelation among the labels. In contrast, panel (b) shows the correlation matrix of the extracted components, where the non-diagonal elements are nearly zero, indicating effective decorrelation.

- Compared to the results reported in [51], where some obtained components remain correlated, the non-diagonal elements in panel (b) are fully eliminated. This difference arises because the Fourier transform in [51] achieves decorrelation only when the original label sequence is nearly infinitely long ($T \rightarrow \infty$), a condition that is not met in real-world applications with finite forecast horizons.

This limitation stems from the predefined nature of the projection matrix, which lacks adaptation to the specific properties of the data. In contrast, our method ensures decorrelation by solving a constrained optimization problem, without relying on an infinitely long forecast horizon, thereby providing a more reliable approach for handling autocorrelation bias.

# B  Theoretical Justification

**Theorem B.1** (Autocorrelation bias, Theorem 3.1 in the main text). *Given a univariate label sequence* $\mathbf{y} \in \mathbb{R}^{\mathrm{T}}$ *where* $\boldsymbol{\Sigma} \in \mathbb{R}^{\mathrm{T} \times \mathrm{T}}$ *denotes the step-wise correlation coefficient, the loss function* $\mathcal{L}_{\mathrm{MSE}}$ *in* (1) *is biased from the true negative log-likelihood of the label sequence, which is given by:*

$$\mathrm{Bias} = \|\mathbf{y} - \hat{\mathbf{y}}\|_{\boldsymbol{\Sigma}^{-1}}^2 - \|\mathbf{y} - \hat{\mathbf{y}}\|^2 - \frac{1}{2}\log|\boldsymbol{\Sigma}|. \tag{6}$$

*where* $\|\mathbf{v}\|_{\boldsymbol{\Sigma}^{-1}}^2 = \mathbf{v}^\top \boldsymbol{\Sigma}^{-1} \mathbf{v}$. *The bias vanishes if different steps in* $\mathbf{y}$ *are decorrelated.*[5]

*Proof.* The proof follows our previous work [51] but relaxes the first-order Markov assumption.

Suppose the label sequence follows a multivariate normal distribution with mean vector $\boldsymbol{\mu} = \hat{\mathbf{y}}$ and covariance matrix $\boldsymbol{\Sigma}$, where the off-diagonal entries are $\Sigma_{ij} = \rho_{ij}\sigma^2$ for $i \neq j$. Here, $\rho_{ij}$ denotes the partial correlation between $y_i$ and $y_j$. The log-likelihood of the label sequence $Y$ can be expressed as:

$$\log p(\mathbf{y}) = \frac{1}{2}\left(\mathrm{T}\log(2\pi) + \log|\boldsymbol{\Sigma}| + (\mathbf{y} - \hat{\mathbf{y}})^\top \boldsymbol{\Sigma}^{-1}(\mathbf{y} - \hat{\mathbf{y}})\right).$$

Removing the constant terms unrelated to $\hat{\mathbf{y}}$, we obtain the practical negative log-likelihood (PNLL):

$$\mathrm{PNLL} = (\mathbf{y} - \hat{\mathbf{y}})^\top \boldsymbol{\Sigma}^{-1}(\mathbf{y} - \hat{\mathbf{y}}).$$

On the other hand, the TMSE loss can be expressed as:

$$\mathrm{TMSE} = \|\mathbf{y} - \hat{\mathbf{y}}\|_2^2 = (\mathbf{y} - \hat{\mathbf{y}})^\top \mathbf{I}^{-1}(\mathbf{y} - \hat{\mathbf{y}}).$$

where $\mathbf{I}$ is the identity matrix. The difference between TMSE and PNLL can be expressed as:

$$\mathrm{Bias} = \mathrm{PNLL} - \mathrm{TMSE} = (\mathbf{y} - \hat{\mathbf{y}})^\top \boldsymbol{\Sigma}^{-1}(\mathbf{y} - \hat{\mathbf{y}}) - (\mathbf{y} - \hat{\mathbf{y}})^\top \mathbf{I}(\mathbf{y} - \hat{\mathbf{y}}),$$

which immediately vanishes if the label sequence is decorrelated, i.e., $\boldsymbol{\Sigma} = \mathbf{I}$. The proof is completed. $\square$

**Lemma B.2** (Lemma 3.3 in the main text). *The projection matrix* $\mathbf{P}^*$ *can be obtained via singular value decomposition (SVD):* $\mathbf{Y} = \mathbf{U}\boldsymbol{\Lambda}(\mathbf{P}^*)^\top$, *where* $\mathbf{U} \in \mathbb{R}^{\mathrm{N} \times \mathrm{N}}$ *and* $\mathbf{P}^* \in \mathbb{R}^{\mathrm{K} \times \mathrm{K}}$ *consist of singular vectors, and the diagonal of* $\boldsymbol{\Lambda} \in \mathbb{R}^{\mathrm{N} \times \mathrm{K}}$ *consists of singular values.*

*Proof.* The proof is available in Section 7.3 in [43].

$\square$

**Lemma B.3** (Decorrelated components, Lemma 3.2 in the main text). *Suppose* $\mathbf{Y} \in \mathbb{R}^{\mathrm{N} \times \mathrm{T}}$ *contains normalized label sequences for* N *samples,* $\mathbf{Z} = [\mathbf{z}_1, ..., \mathbf{z}_{\mathrm{K}}]$ *are the obtained components; for any* $k \neq k'$, *we have* $\mathbf{z}_k^\top \mathbf{z}_{k'} = 0$.

*Proof.* For any two latent components $\mathbf{z}_k$ and $\mathbf{z}_{k'}$ with $k \neq k'$, we have:

$$\mathbf{z}_k^\top \mathbf{z}_{k'} = (\mathbf{Y}\mathbf{p}_k)^\top(\mathbf{Y}\mathbf{p}_{k'}) = \mathbf{p}_k^\top \mathbf{Y}^\top \mathbf{Y}\mathbf{p}_{k'} \tag{7}$$

Noting that $\mathbf{p}_k$ and $\mathbf{p}_{k'}$ are eigenvectors of $\mathbf{Y}^\top \mathbf{Y}$ [43], we have

$$\mathbf{Y}^\top \mathbf{Y}\mathbf{p}_k = \lambda_k \mathbf{p}_k, \qquad \mathbf{Y}^\top \mathbf{Y}\mathbf{p}_{k'} = \lambda_{k'}\mathbf{p}_{k'}, \tag{8}$$

which immediately follows by $\mathbf{z}_k^\top \mathbf{z}_{k'} = \lambda_{k'}\mathbf{p}_k^\top \mathbf{p}_{k'}$. Recalling that different projection bases are constrained to orthogonal, i.e., $\mathbf{p}_k^\top \mathbf{p}_{k'} = 0$ for $k \neq k'$, we have

$$\mathbf{z}_k^\top \mathbf{z}_{k'} = 0 \qquad \text{for all } k \neq k', \tag{9}$$

which completes the proof. $\square$

---

[5]The pioneering work [51] identifies the bias under the first-order Markov assumption on the label sequence. This study generalizes this bias without the first-order Markov assumption.

## C  Generalized Orthogonalization and Decorrelation Methods

In this section, we introduce alternative transforms for obtaining latent components, each with distinct characteristics such as dimensionality reduction and noise isolation. A comparatively empirical study is provided in Section 4.5.

**RPCA.** The robust principal component analysis decomposes the data into a low-rank informative component and a sparse noise component, effectively separating structured signals from noise. Specifically, given $\mathbf{Y} \in \mathbb{R}^{N \times T}$, it is achieved by solving:

$$\min_{\mathbf{V},\mathbf{S}} \|\mathbf{V}\|_* + \lambda \|\mathbf{S}\|_1, \quad \text{subject to} \quad \mathbf{Y} = \mathbf{V} + \mathbf{S}, \tag{10}$$

where $\|\cdot\|_*$ is the nuclear norm, $\|\cdot\|_1$ is the element-wise $\ell_1$ norm, and $\lambda$ is a regularization parameter. Afterwards, it performs the principal component analysis on the obtained informative component $\mathbf{V}$ to derive the projection matrix $\mathbf{P}$. The latent components are generated by $\mathbf{Z} = \mathbf{YP}$. While this approach enhances noise elimination, it does not guarantee decorrelation of the derived components, as the projection matrix $\mathbf{P}$ is derived from $\mathbf{V}$ instead of the original data matrix $\mathbf{Y}$.

**SVD.** The singular value decomposition provides a method to decompose the matrix into different components. Given $\mathbf{Y} \in \mathbb{R}^{N \times T}$, we have:

$$\mathbf{Y} = \mathbf{U}\boldsymbol{\Lambda}\mathbf{V}^\top, \tag{11}$$

where $\mathbf{U} \in \mathbb{R}^{N \times r}$ and $\mathbf{V} \in \mathbb{R}^{T \times R}$ are singular vectors, $\boldsymbol{\Lambda} \in \mathbb{R}^{r \times r}$ is diagonal with rank r. The right singular vector is used as the projection matrix, and the latent components are generated by $\mathbf{Z} = \mathbf{YV}$. One key distinction here needs to be highlighted. Unlike the workflow in the main text (Algorithm 1), the label sequence is not normalized after window generation before computing SVD here, resulting in non-decorrelated components.

**FA.** Factor analysis models the observed data as linear combinations of a small number of latent factors plus noise, capturing the covariance structure through these unobserved factors. Specifically, given mean-centered $\mathbf{Y} \in \mathbb{R}^{N \times T}$, the model assumes:

$$\mathbf{Y} = \mathbf{VF}^\top + \mathbf{E}, \tag{12}$$

where $\mathbf{V} \in \mathbb{R}^{N \times K}$ is the factor loading matrix, K is the number of latent factors ($K \ll N$), $\mathbf{F} \in \mathbb{R}^{T \times K}$ contains the latent factor scores for each sample, and $\mathbf{E} \in \mathbb{R}^{N \times T}$ is the noise matrix. The standard assumption is that each factor $f_i \sim \mathcal{N}(0, \mathbf{I})$ and noise $\epsilon_i \sim \mathcal{N}(0, \Psi)$, where $\Psi$ is a diagonal covariance matrix. The loadings $\mathbf{V}$ and factor scores $\mathbf{F}$ are typically estimated via maximum likelihood. The latent components are given by the estimated factor scores, *i.e.*, $\mathbf{Z} = \mathbf{Y}\Psi^{-1}\mathbf{F}(\mathbf{I} + \mathbf{F}^\top\Psi^{-1}\mathbf{F})^{-1} := \mathbf{YP}$ [6]. This approach captures the covariance structure of $\mathbf{Y}$ via a small number of factors, but does not necessarily guarantee uncorrelated or noise-isolated components.

## D  Complexity Analysis

In this section, we analyze the running cost of Time-o1. The core computation of Time-o1 involves (a) calculating the projection matrix $\mathbf{P}^*$ via SVD, and (b) performing transformation on both $\mathbf{Y}$ and $\hat{\mathbf{Y}}$ followed by calculating their point-wise MAE loss. Given the target matrix $\mathbf{Y} \in \mathbb{R}^{N \times T}$, the SVD step decomposes $\mathbf{Y}$ with an established complexity of $\mathcal{O}(NT^2)$ (assuming $N \geq T$ and $K = T$). For the sequence transformation, each sample (row) in $\mathbf{Y}$ is multiplied by the projection matrix $\mathbf{P}^* \in \mathbb{R}^{T \times T}$, resulting in a total complexity of $\mathcal{O}(NT^2)$. The computation of point-wise MAE loss across all samples and forecast steps is $\mathcal{O}(NT)$, which is negligible compared to the complexity of previous steps. Thus, the overall complexity per batch is dominated by the SVD and projection operations, both scaling as $\mathcal{O}(NT^2)$. The main findings from the empirical evaluations are as follows.

- Fig. 5 (a) presents the computational cost for calculating the projection matrix. Overall, it increases linearly with the sample size and quadratically with the prediction length, which aligns with the theoretical complexity. Importantly, this operation is performed only once before training begins, rendering the associated overhead acceptable.

---

[6]Adapted from source code of sklearn: https://github.com/scikit-learn/scikit-learn/blob/98ed9dc73/sklearn/decomposition/

- Fig. 5 (b) presents the computational cost for the sequence transformation. The cost increases quadratically with the prediction length, but remains below 2 ms. This cost is comparable to that of a linear projection. Furthermore, sequence transformation is not required during inference.

*In conclusion, Time-o1 does not add complexity to model inference, and the additional complexity during the training stage is negligible.*

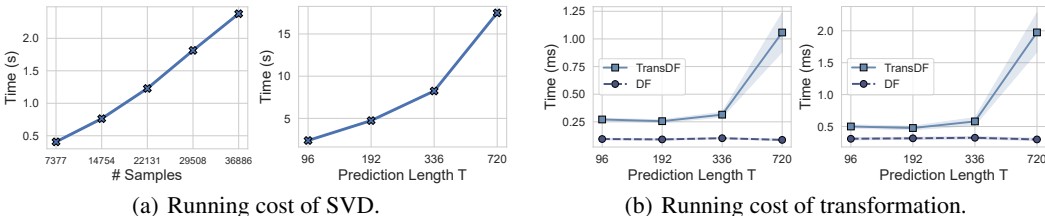

      (a) Running cost of SVD.           (b) Running cost of transformation.

Figure 5: Running cost for projection matrix calculation (left panel with varying number of samples, right panel with varying prediction length) and sequence transformation (left panel for forward pass, right panel for backward pass, with average and shaded areas for 95% confidence intervals).

# E    More Experimental Results

## E.1    Long-term forecast performance

Additional results on long-term forecast performance are available in Table 7.

## E.2    Short-term forecast performance

Additional results on short-term forecast performance are available in Table 8, where Fredformer [37] serves as the forecast model.

## E.3    Showcases

Additional results on showcases are available in Fig. 6 and Fig. 7.

## E.4    Generalization studies

Additional results on varying forecast models and transformations are available in Fig. 8 and Table 9.

## E.5    Hyperparameter sensitivity

Additional results on hyperparameter sensitivity are available in Fig. 9 for $\alpha$ and Fig. 10 for $\gamma$.

## E.6    Comparison with different loss functions

Additional results on comparing different loss functions are available in Table 10.

## E.7    Varying input length results

Additional results on varying input lengths are available in Table 11—complementing the fixed length of 96 used in the main text.

## E.8    Random seed sensitivity

Additional results on varying random seeds are available in Table 12.

Table 7: The comprehensive results on the long-term forecasting task.

| Models | | Time-o1 (Ours) | | Fredformer (2024) | | iTransformer (2024) | | FreTS (2023) | | TimesNet (2023) | | MICN (2023) | | TiDE (2023) | | DLinear (2023) | | FEDformer (2022) | | Autoformer (2021) | | Transformer (2017) | |
|---|---|---|---|---|---|---|---|---|---|---|---|---|---|---|---|---|---|---|---|---|---|---|---|
| Metrics | | MSE | MAE | MSE | MAE | MSE | MAE | MSE | MAE | MSE | MAE | MSE | MAE | MSE | MAE | MSE | MAE | MSE | MAE | MSE | MAE | MSE | MAE |
| ETTm1 | 96 | **0.321** | **0.357** | 0.326 | 0.361 | 0.338 | 0.372 | 0.342 | 0.375 | 0.368 | 0.394 | **0.319** | 0.366 | 0.353 | 0.374 | 0.346 | 0.373 | 0.401 | 0.434 | 0.485 | 0.468 | 0.503 | 0.482 |
| | 192 | **0.360** | **0.378** | 0.365 | 0.382 | 0.382 | 0.396 | 0.385 | 0.400 | 0.406 | 0.409 | 0.364 | 0.395 | 0.391 | 0.393 | 0.380 | 0.390 | 0.415 | 0.446 | 0.504 | 0.482 | 0.807 | 0.664 |
| | 336 | **0.389** | **0.400** | 0.396 | 0.404 | 0.427 | 0.424 | 0.416 | 0.421 | 0.454 | 0.444 | 0.395 | 0.425 | 0.423 | 0.414 | 0.413 | 0.414 | 0.432 | 0.450 | 0.520 | 0.489 | 0.847 | 0.678 |
| | 720 | **0.447** | **0.435** | 0.459 | 0.444 | 0.496 | 0.463 | 0.513 | 0.489 | 0.527 | 0.474 | 0.505 | 0.499 | 0.486 | 0.448 | 0.472 | 0.450 | 0.522 | 0.500 | 0.594 | 0.523 | 1.037 | 0.771 |
| | Avg | **0.379** | **0.393** | 0.387 | 0.398 | 0.411 | 0.414 | 0.414 | 0.421 | 0.438 | 0.430 | 0.396 | 0.421 | 0.413 | 0.407 | 0.403 | 0.407 | 0.442 | 0.457 | 0.526 | 0.491 | 0.799 | 0.648 |
| ETTm2 | 96 | **0.172** | **0.251** | 0.177 | 0.260 | 0.182 | 0.265 | 0.188 | 0.279 | 0.184 | 0.262 | 0.178 | 0.277 | 0.182 | 0.265 | 0.188 | 0.283 | 0.205 | 0.289 | 0.218 | 0.300 | 0.386 | 0.441 |
| | 192 | **0.235** | **0.294** | 0.242 | 0.300 | 0.257 | 0.315 | 0.264 | 0.329 | 0.257 | 0.308 | 0.266 | 0.343 | 0.247 | 0.304 | 0.280 | 0.356 | 0.271 | 0.334 | 0.282 | 0.340 | 1.410 | 0.881 |
| | 336 | **0.293** | **0.333** | 0.302 | 0.340 | 0.320 | 0.354 | 0.322 | 0.369 | 0.315 | 0.345 | 0.299 | 0.354 | 0.307 | 0.343 | 0.375 | 0.420 | 0.327 | 0.366 | 0.335 | 0.370 | 1.940 | 1.070 |
| | 720 | **0.388** | **0.389** | 0.399 | 0.397 | 0.423 | 0.411 | 0.489 | 0.482 | 0.452 | 0.421 | 0.489 | 0.482 | 0.408 | 0.398 | 0.526 | 0.508 | 0.428 | 0.425 | 0.423 | 0.420 | 2.914 | 1.276 |
| | Avg | **0.272** | **0.317** | 0.280 | 0.324 | 0.295 | 0.336 | 0.316 | 0.365 | 0.302 | 0.334 | 0.308 | 0.364 | 0.286 | 0.328 | 0.342 | 0.392 | 0.308 | 0.354 | 0.315 | 0.358 | 1.662 | 0.917 |
| ETTh1 | 96 | **0.368** | **0.391** | 0.377 | 0.396 | 0.385 | 0.405 | 0.398 | 0.409 | 0.399 | 0.418 | 0.381 | 0.416 | 0.387 | 0.395 | 0.389 | 0.404 | 0.391 | 0.433 | 0.449 | 0.465 | 1.028 | 0.778 |
| | 192 | **0.424** | **0.422** | 0.437 | 0.425 | 0.440 | 0.437 | 0.451 | 0.442 | 0.452 | 0.451 | 0.497 | 0.489 | 0.439 | 0.425 | 0.442 | 0.440 | 0.418 | 0.448 | 0.459 | 0.469 | 1.010 | 0.789 |
| | 336 | **0.467** | **0.441** | 0.486 | 0.449 | 0.480 | 0.457 | 0.501 | 0.472 | 0.488 | 0.469 | 0.589 | 0.555 | 0.482 | 0.447 | 0.488 | 0.467 | 0.487 | 0.484 | 0.511 | 0.500 | 0.908 | 0.743 |
| | 720 | **0.465** | **0.463** | 0.488 | 0.467 | 0.504 | 0.492 | 0.608 | 0.571 | 0.549 | 0.515 | 0.665 | 0.617 | 0.484 | 0.471 | 0.505 | 0.502 | 0.494 | 0.514 | 0.488 | 0.498 | 0.987 | 0.785 |
| | Avg | **0.431** | **0.429** | 0.447 | 0.434 | 0.452 | 0.448 | 0.489 | 0.474 | 0.472 | 0.463 | 0.533 | 0.519 | 0.448 | 0.435 | 0.456 | 0.453 | 0.447 | 0.470 | 0.477 | 0.483 | 0.983 | 0.774 |
| ETTh2 | 96 | **0.282** | **0.330** | 0.293 | 0.344 | 0.301 | 0.349 | 0.315 | 0.374 | 0.321 | 0.358 | 0.351 | 0.398 | 0.291 | 0.340 | 0.330 | 0.383 | 0.351 | 0.391 | 0.355 | 0.397 | 1.485 | 0.959 |
| | 192 | **0.359** | **0.381** | 0.372 | 0.391 | 0.383 | 0.397 | 0.466 | 0.467 | 0.418 | 0.417 | 0.492 | 0.489 | 0.376 | 0.392 | 0.439 | 0.450 | 0.456 | 0.456 | 0.478 | 0.471 | 4.218 | 1.585 |
| | 336 | **0.394** | **0.414** | 0.420 | 0.433 | 0.425 | 0.432 | 0.522 | 0.502 | 0.464 | 0.454 | 0.656 | 0.582 | 0.417 | 0.427 | 0.589 | 0.538 | 0.477 | 0.492 | 0.459 | 0.469 | 2.775 | 1.361 |
| | 720 | **0.400** | **0.427** | 0.421 | 0.439 | 0.436 | 0.448 | 0.792 | 0.643 | 0.434 | 0.450 | 0.981 | 0.718 | 0.429 | 0.446 | 0.757 | 0.626 | 0.522 | 0.505 | 0.499 | 0.502 | 2.274 | 1.257 |
| | Avg | **0.359** | **0.388** | 0.377 | 0.402 | 0.386 | 0.407 | 0.524 | 0.496 | 0.409 | 0.420 | 0.620 | 0.546 | 0.378 | 0.401 | 0.529 | 0.499 | 0.452 | 0.461 | 0.448 | 0.460 | 2.688 | 1.291 |
| ECL | 96 | **0.145** | **0.235** | 0.161 | 0.258 | 0.150 | 0.242 | 0.180 | 0.266 | 0.170 | 0.272 | 0.170 | 0.281 | 0.197 | 0.274 | 0.197 | 0.282 | 0.187 | 0.302 | 0.189 | 0.304 | 0.253 | 0.350 |
| | 192 | **0.159** | **0.249** | 0.174 | 0.269 | 0.168 | 0.259 | 0.184 | 0.272 | 0.183 | 0.282 | 0.185 | 0.297 | 0.197 | 0.277 | 0.197 | 0.286 | 0.207 | 0.322 | 0.271 | 0.371 | 0.262 | 0.356 |
| | 336 | **0.173** | **0.264** | 0.194 | 0.290 | 0.182 | 0.274 | 0.199 | 0.290 | 0.203 | 0.302 | 0.190 | 0.298 | 0.212 | 0.292 | 0.209 | 0.301 | 0.211 | 0.326 | 0.243 | 0.352 | 0.269 | 0.363 |
| | 720 | **0.203** | **0.292** | 0.235 | 0.319 | 0.214 | 0.304 | 0.234 | 0.322 | 0.294 | 0.366 | 0.221 | 0.329 | 0.254 | 0.325 | 0.245 | 0.334 | 0.253 | 0.361 | 0.295 | 0.388 | 0.277 | 0.365 |
| | Avg | **0.170** | **0.260** | 0.191 | 0.284 | 0.179 | 0.270 | 0.199 | 0.288 | 0.212 | 0.306 | 0.192 | 0.302 | 0.215 | 0.292 | 0.212 | 0.301 | 0.214 | 0.328 | 0.249 | 0.354 | 0.265 | 0.358 |
| Traffic | 96 | **0.393** | **0.265** | 0.461 | 0.327 | 0.397 | 0.271 | 0.531 | 0.323 | 0.590 | 0.316 | 0.498 | 0.298 | 0.646 | 0.386 | 0.649 | 0.397 | 0.588 | 0.367 | 0.575 | 0.356 | 0.689 | 0.396 |
| | 192 | **0.410** | **0.275** | 0.470 | 0.326 | 0.416 | 0.279 | 0.519 | 0.321 | 0.624 | 0.336 | 0.521 | 0.309 | 0.599 | 0.362 | 0.598 | 0.371 | 0.613 | 0.377 | 0.647 | 0.394 | 0.710 | 0.388 |
| | 336 | **0.421** | **0.280** | 0.492 | 0.338 | 0.429 | 0.286 | 0.529 | 0.327 | 0.641 | 0.345 | 0.529 | 0.314 | 0.606 | 0.363 | 0.605 | 0.373 | 0.640 | 0.398 | 0.694 | 0.446 | 0.687 | 0.366 |
| | 720 | **0.451** | **0.298** | 0.521 | 0.353 | 0.462 | 0.303 | 0.573 | 0.346 | 0.670 | 0.356 | 0.567 | 0.326 | 0.643 | 0.383 | 0.646 | 0.395 | 0.718 | 0.450 | 0.731 | 0.468 | 0.681 | 0.366 |
| | Avg | **0.419** | **0.280** | 0.486 | 0.336 | 0.426 | 0.285 | 0.538 | 0.330 | 0.631 | 0.338 | 0.529 | 0.312 | 0.624 | 0.373 | 0.625 | 0.384 | 0.640 | 0.398 | 0.662 | 0.416 | 0.692 | 0.379 |
| Weather | 96 | **0.169** | 0.219 | 0.180 | 0.220 | 0.171 | **0.210** | 0.174 | 0.228 | 0.183 | 0.229 | 0.179 | 0.244 | 0.192 | 0.232 | 0.194 | 0.253 | 0.235 | 0.310 | 0.233 | 0.306 | 0.423 | 0.448 |
| | 192 | **0.210** | **0.258** | 0.222 | 0.258 | 0.246 | 0.278 | 0.213 | 0.266 | 0.242 | 0.276 | 0.242 | 0.310 | 0.240 | 0.270 | 0.238 | 0.296 | 0.295 | 0.353 | 0.286 | 0.347 | 0.664 | 0.585 |
| | 336 | **0.259** | **0.297** | 0.283 | 0.301 | 0.296 | 0.313 | 0.270 | 0.316 | 0.293 | 0.312 | 0.273 | 0.330 | 0.292 | 0.307 | 0.282 | 0.332 | 0.364 | 0.397 | 0.346 | 0.385 | 0.848 | 0.686 |
| | 720 | **0.327** | 0.349 | 0.358 | **0.348** | 0.362 | 0.353 | 0.337 | 0.362 | 0.366 | 0.361 | 0.360 | 0.399 | 0.364 | 0.353 | 0.347 | 0.385 | 0.411 | 0.429 | 0.412 | 0.420 | 0.861 | 0.685 |
| | Avg | **0.241** | **0.280** | 0.261 | 0.282 | 0.269 | 0.289 | 0.249 | 0.293 | 0.271 | 0.295 | 0.264 | 0.321 | 0.272 | 0.291 | 0.265 | 0.317 | 0.326 | 0.372 | 0.319 | 0.365 | 0.699 | 0.601 |
| PEMS03 | 12 | **0.070** | **0.176** | 0.081 | 0.191 | 0.072 | 0.179 | 0.085 | 0.198 | 0.094 | 0.201 | 0.096 | 0.217 | 0.117 | 0.226 | 0.105 | 0.220 | 0.108 | 0.229 | 0.233 | 0.366 | 0.106 | 0.206 |
| | 24 | **0.087** | **0.198** | 0.121 | 0.240 | 0.104 | 0.217 | 0.129 | 0.244 | 0.116 | 0.221 | 0.095 | 0.210 | 0.233 | 0.322 | 0.183 | 0.297 | 0.131 | 0.255 | 0.405 | 0.485 | 0.117 | 0.221 |
| | 36 | **0.105** | **0.219** | 0.180 | 0.292 | 0.137 | 0.251 | 0.173 | 0.286 | 0.134 | 0.237 | 0.107 | 0.223 | 0.379 | 0.418 | 0.258 | 0.361 | 0.159 | 0.285 | 0.327 | 0.415 | 0.127 | 0.233 |
| | 48 | **0.124** | **0.238** | 0.201 | 0.316 | 0.174 | 0.285 | 0.207 | 0.315 | 0.161 | 0.262 | 0.125 | 0.242 | 0.535 | 0.516 | 0.319 | 0.410 | 0.209 | 0.331 | 0.679 | 0.634 | 0.139 | 0.245 |
| | Avg | **0.097** | **0.208** | 0.146 | 0.260 | 0.122 | 0.233 | 0.149 | 0.261 | 0.126 | 0.230 | 0.106 | 0.223 | 0.316 | 0.370 | 0.216 | 0.322 | 0.152 | 0.275 | 0.411 | 0.475 | 0.122 | 0.226 |
| PEMS08 | 12 | **0.081** | **0.183** | 0.091 | 0.199 | 0.084 | 0.187 | 0.096 | 0.205 | 0.111 | 0.208 | 0.161 | 0.274 | 0.121 | 0.233 | 0.113 | 0.225 | 0.163 | 0.258 | 0.232 | 0.334 | 0.204 | 0.232 |
| | 24 | **0.117** | **0.218** | 0.138 | 0.245 | 0.123 | 0.227 | 0.151 | 0.258 | 0.139 | 0.232 | 0.127 | 0.237 | 0.232 | 0.325 | 0.199 | 0.302 | 0.197 | 0.288 | 0.545 | 0.550 | 0.232 | 0.251 |
| | 36 | **0.157** | **0.253** | 0.199 | 0.303 | 0.170 | 0.268 | 0.203 | 0.303 | 0.168 | 0.260 | 0.148 | 0.252 | 0.376 | 0.427 | 0.295 | 0.371 | 0.241 | 0.326 | 0.379 | 0.436 | 0.246 | 0.263 |
| | 48 | 0.207 | 0.294 | 0.255 | 0.338 | 0.218 | 0.306 | 0.247 | 0.334 | **0.189** | **0.272** | 0.175 | 0.270 | 0.543 | 0.527 | 0.389 | 0.429 | 0.302 | 0.375 | 0.531 | 0.502 | 0.278 | 0.297 |
| | Avg | **0.141** | **0.237** | 0.171 | 0.271 | 0.149 | 0.247 | 0.174 | 0.275 | 0.152 | 0.243 | 0.153 | 0.258 | 0.318 | 0.378 | 0.249 | 0.332 | 0.226 | 0.312 | 0.422 | 0.456 | 0.240 | 0.261 |
| 1st Count | | **43** | **42** | 0 | 1 | 0 | 1 | 0 | 0 | 1 | 1 | 1 | 0 | 0 | 0 | 0 | 0 | 0 | 0 | 0 | 0 | 0 | 0 |

*Note*: We fix the input length as 96 following [28]. **Bold** typeface highlights the top performance for each metric, while underlined text denotes the second-best results. *Avg* indicates the results averaged over forecasting lengths: T=96, 192, 336 and 720.

Table 8: The comprehensive results on the short-term forecasting task.

| Models | Time-o1 (Ours) | | | Fredformer (2024) | | | iTransformer (2024) | | | FreTS (2023) | | | MICN (2023) | | | DLinear (2023) | | | Fedformer (2023) | | |
|---|---|---|---|---|---|---|---|---|---|---|---|---|---|---|---|---|---|---|---|---|---|
| Metric | SMAPE | MASE | OWA | SMAPE | MASE | OWA | SMAPE | MASE | OWA | SMAPE | MASE | OWA | SMAPE | MASE | OWA | SMAPE | MASE | OWA | SMAPE | MASE | OWA |
| Yearly | **13.485** | **3.010** | **0.791** | 13.509 | 3.028 | 0.794 | 13.797 | 3.143 | 0.818 | 13.576 | 3.068 | 0.801 | 14.594 | 3.392 | 0.873 | 14.307 | 3.094 | 0.827 | 13.648 | 3.089 | 0.806 |
| Quarterly | **10.105** | **1.180** | **0.889** | 10.140 | 1.185 | 0.893 | 10.503 | 1.248 | 0.932 | 10.361 | 1.223 | 0.916 | 11.417 | 1.385 | 1.023 | 10.500 | 1.237 | 0.928 | 10.612 | 1.246 | 0.936 |
| Monthly | **12.649** | **0.930** | **0.875** | 12.696 | 0.931 | 0.878 | 13.227 | 1.013 | 0.935 | 13.088 | 0.990 | 0.919 | 13.834 | 1.080 | 0.987 | 13.362 | 1.007 | 0.937 | 14.181 | 1.105 | 1.011 |
| Others | 4.852 | 3.274 | 1.027 | 4.848 | **3.230** | **1.019** | 5.101 | 3.419 | 1.076 | 5.563 | 3.71 | 1.17 | 6.137 | 4.201 | 1.308 | 5.12 | 3.649 | 1.114 | **4.823** | 3.243 | 1.019 |
| Average | **11.841** | **1.585** | **0.851** | 11.879 | 1.590 | 0.854 | 12.298 | 1.68 | 0.893 | 12.169 | 1.66 | 0.883 | 13.044 | 1.841 | 0.962 | 12.48 | 1.674 | 0.898 | 12.734 | 1.702 | 0.914 |
| 1st Count | **4** | **4** | **4** | 0 | 1 | 1 | 0 | 0 | 0 | 0 | 0 | 0 | 0 | 0 | 0 | 0 | 0 | 0 | 1 | 0 | 0 |

*Note*: **Bold** typeface highlights the top performance for each metric, while underlined text denotes the second-best results. *Avg* indicates the results averaged over forecasting lengths: yearly, quarterly, and monthly.

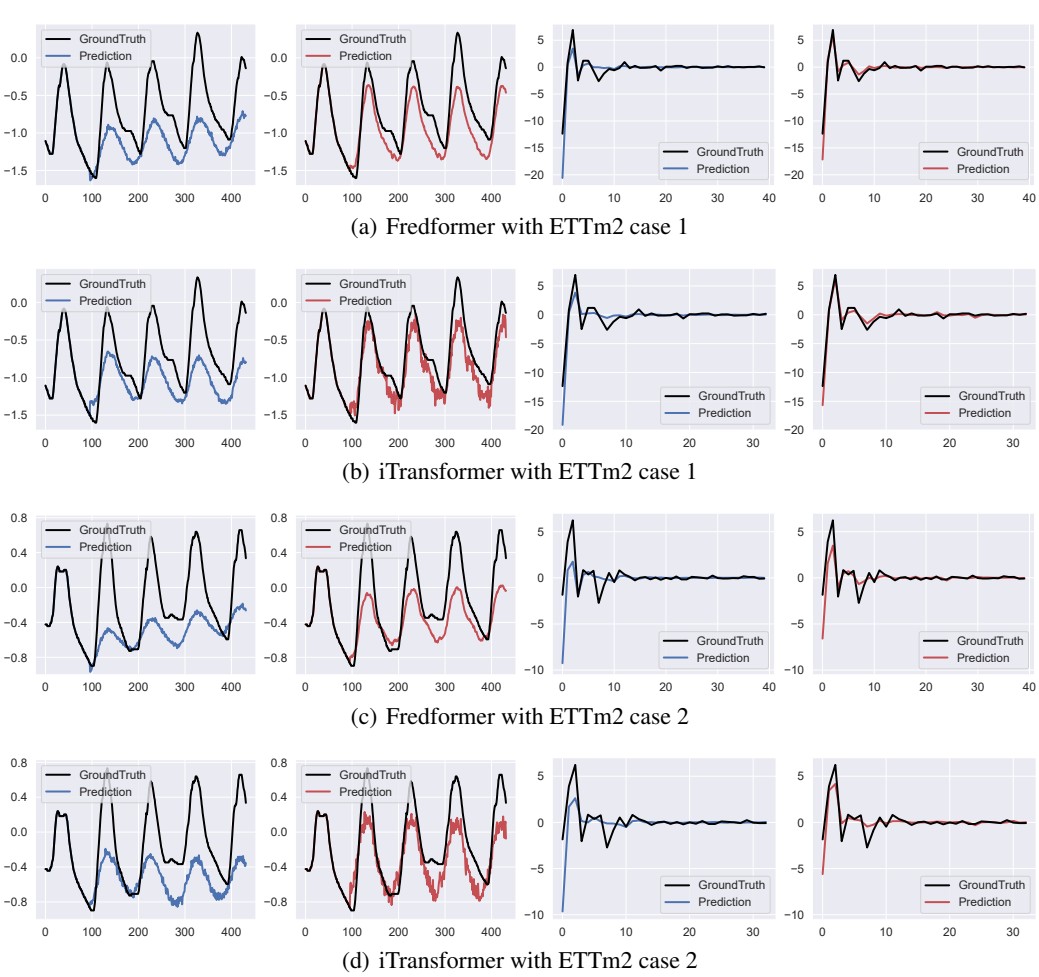

(a) Fredformer with ETTm2 case 1

(b) iTransformer with ETTm2 case 1

(c) Fredformer with ETTm2 case 2

(d) iTransformer with ETTm2 case 2

Figure 6: The forecast sequences generated with DF and Time-o1. The forecast length is set to 336 and the experiment is conducted on ETTm2.

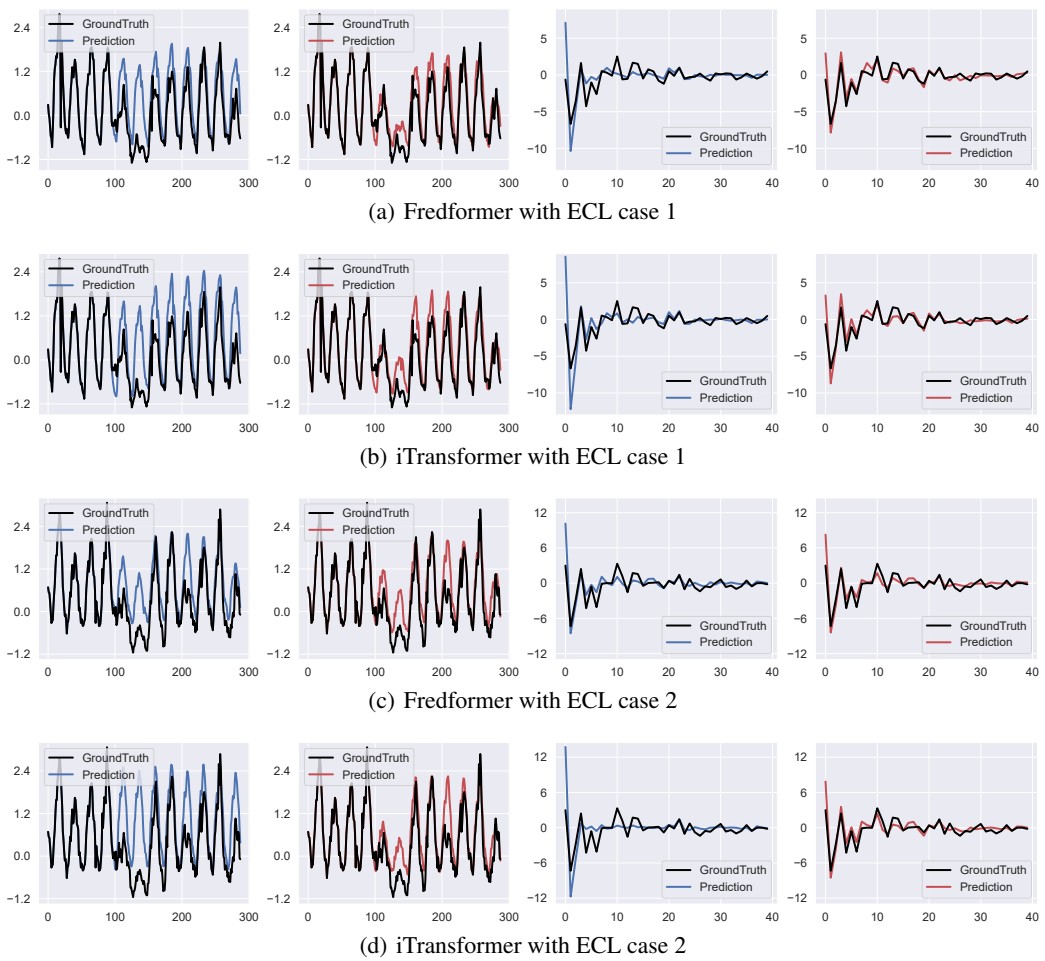

Figure 7: The forecast sequences generated with DF and Time-o1. The forecast length is set to 192 and the experiment is conducted on ECL.

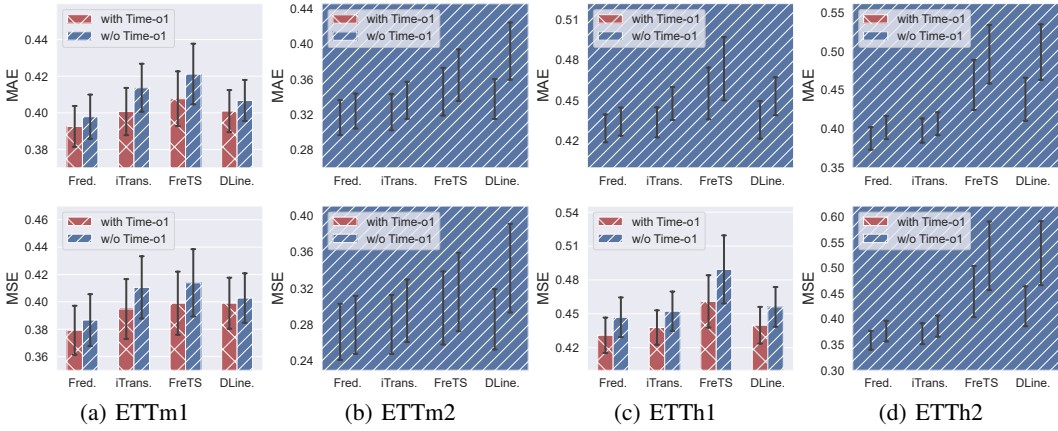

Figure 8: Performance of different forecast models with and without Time-o1. The forecast errors are averaged over forecast lengths and the error bars represent 50% confidence intervals.

Table 9: Varying transformation results.

| Trans | | PCA | | RPCA | | SVD | | FA | | DF | |
|---|---|---|---|---|---|---|---|---|---|---|---|
| Metrics | | MSE | MAE | MSE | MAE | MSE | MAE | MSE | MAE | MSE | MAE |
| ECL | 96 | **0.1449** | **0.2348** | 0.1450 | 0.2349 | 0.1450 | 0.2350 | 0.1478 | 0.2385 | 0.1500 | 0.2415 |
| | 192 | **0.1592** | **0.2487** | 0.1594 | 0.2487 | 0.1595 | 0.2490 | 0.1619 | 0.2517 | 0.1681 | 0.2591 |
| | 336 | 0.1731 | 0.2645 | 0.1732 | 0.2646 | **0.1730** | **0.2643** | 0.1789 | 0.2711 | 0.1823 | 0.2744 |
| | 720 | **0.2033** | **0.2920** | 0.2066 | 0.2960 | 0.2214 | 0.3066 | 0.2095 | 0.2975 | 0.2145 | 0.3035 |
| | Avg | **0.1701** | **0.2600** | 0.1710 | 0.2611 | 0.1747 | 0.2637 | 0.1745 | 0.2647 | 0.1787 | 0.2696 |
| Weather | 96 | **0.1692** | **0.2185** | 0.1715 | 0.2199 | 0.1723 | 0.2223 | 0.1717 | 0.2247 | 0.1737 | 0.2277 |
| | 192 | **0.2102** | **0.2575** | 0.2116 | 0.2590 | 0.2116 | 0.2597 | 0.2125 | 0.2636 | 0.2128 | 0.2661 |
| | 336 | **0.2586** | **0.2971** | 0.2631 | 0.3072 | 0.2676 | 0.3110 | 0.2613 | 0.2997 | 0.2705 | 0.3159 |
| | 720 | **0.3271** | **0.3487** | 0.3303 | 0.3581 | 0.3394 | 0.3681 | 0.3354 | 0.3610 | 0.3372 | 0.3623 |
| | Avg | **0.2413** | **0.2805** | 0.2441 | 0.2860 | 0.2477 | 0.2903 | 0.2452 | 0.2872 | 0.2486 | 0.2930 |

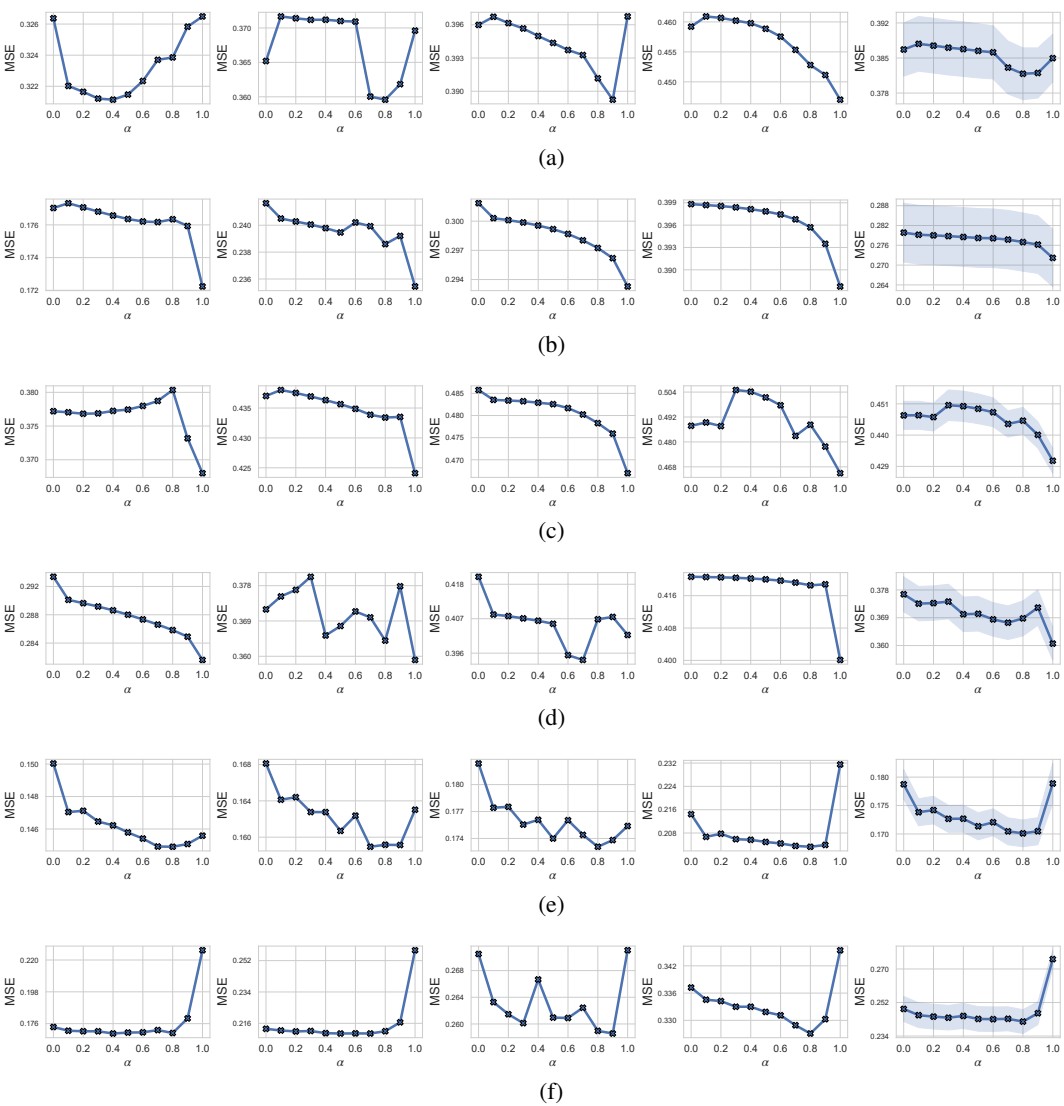

Figure 9: Time-o1 improves Fredformer performance given a wide range of transformed loss strength $\alpha$. These experiments are conducted on ETTh1 (a), ETTh2 (b), ETTm1 (c), ETTm2 (d), ECL (e), Weather (f) datasets. Different columns correspond to different forecast lengths (from left to right: 96, 192, 336, 720, and their average with shaded areas being 15% confidence intervals).

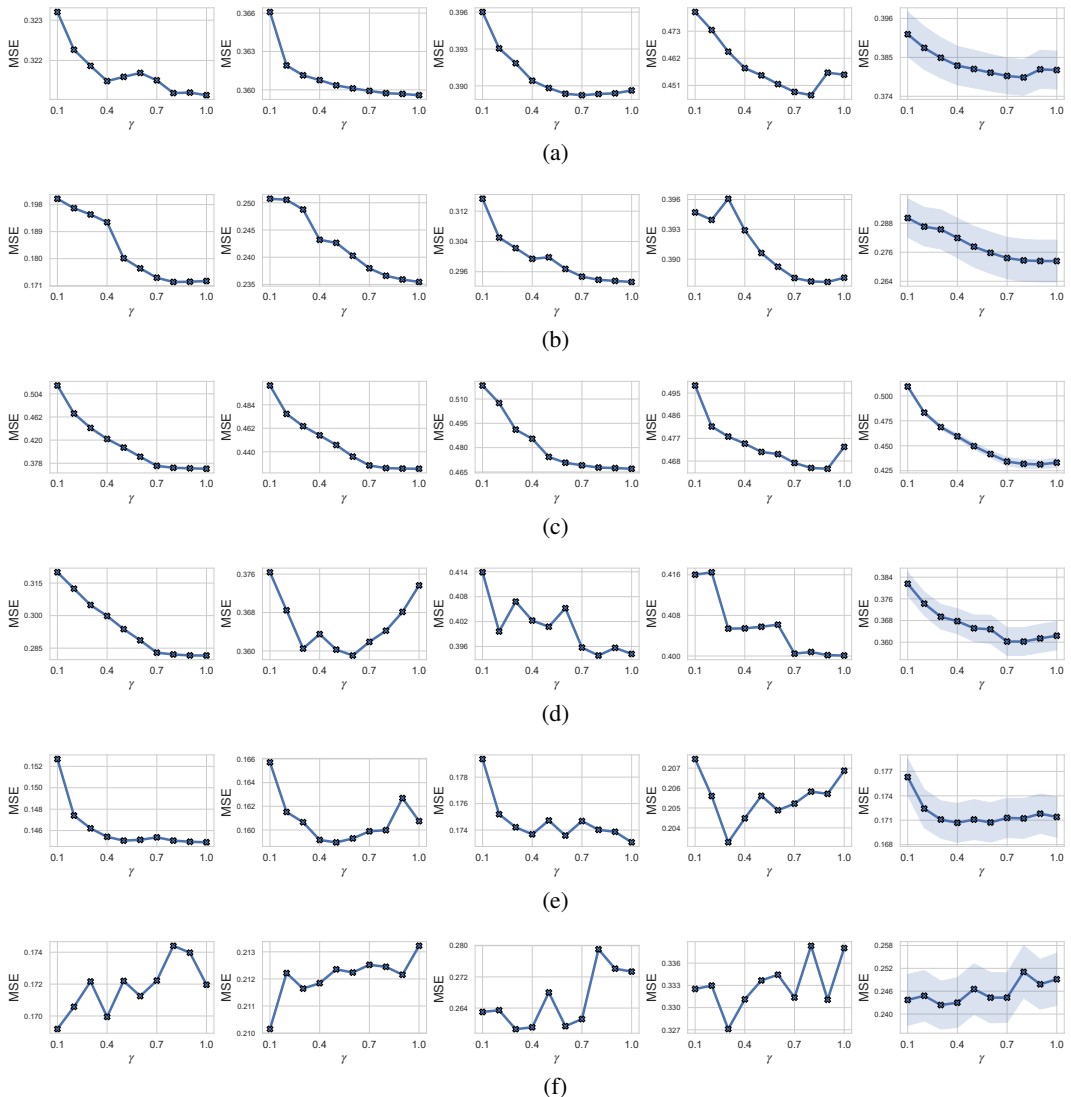

Figure 10: Time-o1 improves Fredformer performance given a wide range of rank ratio $\gamma$. These experiments are conducted on ETTh1 (a), ETTh2 (b), ETTm1 (c), ETTm2 (d), ECL (e), and Weather (f) datasets. Different columns correspond to different forecast lengths (from left to right: 96, 192, 336, 720, and their average with shaded areas being 15% confidence intervals).

Table 10: Comparable results with different loss functions.

| Loss | | **Time-o1** | | FreDF | | Koopman | | Dilate | | Soft-DTW | | DPTA | | DF | |
|---|---|---|---|---|---|---|---|---|---|---|---|---|---|---|---|
| Metrics | | MSE | MAE | MSE | MAE | MSE | MAE | MSE | MAE | MSE | MAE | MSE | MAE | MSE | MAE |
| **Forecast model: FredFormer** | | | | | | | | | | | | | | | |
| ETTm1 | 96 | 0.321 | 0.357 | 0.326 | 0.355 | 0.335 | 0.368 | 0.337 | 0.367 | 0.332 | 0.363 | 0.332 | 0.364 | 0.326 | 0.361 |
| | 192 | 0.360 | 0.378 | 0.363 | 0.380 | 0.366 | 0.384 | 0.364 | 0.384 | 0.370 | 0.386 | 0.370 | 0.386 | 0.365 | 0.382 |
| | 336 | 0.389 | 0.400 | 0.392 | 0.400 | 0.399 | 0.408 | 0.397 | 0.406 | 0.406 | 0.409 | 0.409 | 0.410 | 0.396 | 0.404 |
| | 720 | 0.447 | 0.435 | 0.455 | 0.440 | 0.456 | 0.441 | 0.457 | 0.443 | 0.478 | 0.450 | 0.476 | 0.448 | 0.459 | 0.444 |
| | Avg | 0.379 | 0.393 | 0.384 | 0.394 | 0.389 | 0.400 | 0.389 | 0.400 | 0.397 | 0.402 | 0.396 | 0.402 | 0.387 | 0.398 |
| ETTh1 | 96 | 0.368 | 0.391 | 0.370 | 0.392 | 0.375 | 0.397 | 0.378 | 0.399 | 0.376 | 0.398 | 0.378 | 0.399 | 0.377 | 0.396 |
| | 192 | 0.424 | 0.422 | 0.436 | 0.437 | 0.438 | 0.434 | 0.439 | 0.435 | 0.439 | 0.435 | 0.438 | 0.433 | 0.437 | 0.425 |
| | 336 | 0.467 | 0.441 | 0.473 | 0.443 | 0.473 | 0.455 | 0.481 | 0.453 | 0.484 | 0.455 | 0.486 | 0.455 | 0.486 | 0.449 |
| | 720 | 0.465 | 0.463 | 0.474 | 0.466 | 0.523 | 0.487 | 0.516 | 0.482 | 0.542 | 0.510 | 0.538 | 0.510 | 0.488 | 0.467 |
| | Avg | 0.431 | 0.429 | 0.438 | 0.434 | 0.452 | 0.443 | 0.453 | 0.442 | 0.460 | 0.449 | 0.460 | 0.449 | 0.447 | 0.434 |
| ECL | 96 | 0.151 | 0.245 | 0.152 | 0.247 | 0.166 | 0.263 | 0.158 | 0.253 | 0.168 | 0.266 | 0.158 | 0.253 | 0.161 | 0.258 |
| | 192 | 0.166 | 0.256 | 0.166 | 0.257 | 0.174 | 0.267 | 0.170 | 0.263 | 0.218 | 0.313 | 0.216 | 0.307 | 0.174 | 0.269 |
| | 336 | 0.181 | 0.274 | 0.183 | 0.278 | 0.188 | 0.280 | 0.190 | 0.286 | 0.197 | 0.291 | 0.199 | 0.295 | 0.194 | 0.290 |
| | 720 | 0.213 | 0.304 | 0.216 | 0.304 | 0.232 | 0.318 | 0.229 | 0.316 | 0.240 | 0.322 | 0.235 | 0.322 | 0.235 | 0.319 |
| | Avg | 0.178 | 0.270 | 0.179 | 0.272 | 0.190 | 0.282 | 0.187 | 0.280 | 0.206 | 0.298 | 0.202 | 0.294 | 0.191 | 0.284 |
| Weather | 96 | 0.171 | 0.208 | 0.174 | 0.213 | 0.174 | 0.214 | 0.173 | 0.214 | 0.173 | 0.213 | 0.179 | 0.219 | 0.180 | 0.220 |
| | 192 | 0.219 | 0.253 | 0.219 | 0.254 | 0.220 | 0.256 | 0.225 | 0.260 | 0.220 | 0.255 | 0.223 | 0.257 | 0.222 | 0.258 |
| | 336 | 0.277 | 0.295 | 0.278 | 0.296 | 0.280 | 0.298 | 0.280 | 0.299 | 0.281 | 0.296 | 0.281 | 0.298 | 0.283 | 0.301 |
| | 720 | 0.353 | 0.346 | 0.354 | 0.347 | 0.354 | 0.347 | 0.355 | 0.348 | 0.369 | 0.355 | 0.356 | 0.347 | 0.358 | 0.348 |
| | Avg | 0.255 | 0.276 | 0.256 | 0.277 | 0.257 | 0.279 | 0.258 | 0.280 | 0.261 | 0.280 | 0.260 | 0.280 | 0.261 | 0.282 |
| **Forecast model: iTransformer** | | | | | | | | | | | | | | | |
| ETTm1 | 96 | 0.323 | 0.358 | 0.334 | 0.365 | 0.350 | 0.382 | 0.342 | 0.376 | 0.339 | 0.373 | 0.341 | 0.375 | 0.338 | 0.372 |
| | 192 | 0.371 | 0.388 | 0.381 | 0.390 | 0.389 | 0.400 | 0.381 | 0.396 | 0.383 | 0.395 | 0.383 | 0.395 | 0.382 | 0.396 |
| | 336 | 0.408 | 0.407 | 0.417 | 0.412 | 0.425 | 0.423 | 0.418 | 0.418 | 0.429 | 0.423 | 0.429 | 0.423 | 0.427 | 0.424 |
| | 720 | 0.477 | 0.450 | 0.489 | 0.453 | 0.489 | 0.458 | 0.487 | 0.457 | 0.516 | 0.469 | 0.512 | 0.467 | 0.496 | 0.463 |
| | Avg | 0.395 | 0.401 | 0.405 | 0.405 | 0.413 | 0.416 | 0.407 | 0.412 | 0.417 | 0.415 | 0.416 | 0.415 | 0.411 | 0.414 |
| ETTh1 | 96 | 0.378 | 0.393 | 0.378 | 0.395 | 0.392 | 0.411 | 0.385 | 0.405 | 0.387 | 0.405 | 0.386 | 0.405 | 0.385 | 0.405 |
| | 192 | 0.428 | 0.423 | 0.428 | 0.423 | 0.446 | 0.442 | 0.440 | 0.437 | 0.443 | 0.439 | 0.441 | 0.439 | 0.440 | 0.437 |
| | 336 | 0.473 | 0.450 | 0.470 | 0.447 | 0.483 | 0.461 | 0.480 | 0.457 | 0.494 | 0.464 | 0.489 | 0.462 | 0.480 | 0.457 |
| | 720 | 0.473 | 0.469 | 0.490 | 0.484 | 0.501 | 0.491 | 0.504 | 0.492 | 0.557 | 0.520 | 0.538 | 0.509 | 0.504 | 0.492 |
| | Avg | 0.438 | 0.434 | 0.442 | 0.437 | 0.455 | 0.451 | 0.452 | 0.448 | 0.470 | 0.457 | 0.463 | 0.454 | 0.452 | 0.448 |
| ECL | 96 | 0.145 | 0.235 | 0.149 | 0.238 | 0.151 | 0.243 | 0.150 | 0.241 | 0.149 | 0.241 | 0.149 | 0.240 | 0.150 | 0.242 |
| | 192 | 0.159 | 0.249 | 0.163 | 0.251 | 0.167 | 0.257 | 0.168 | 0.259 | 0.164 | 0.255 | 0.166 | 0.257 | 0.168 | 0.259 |
| | 336 | 0.173 | 0.264 | 0.179 | 0.268 | 0.182 | 0.275 | 0.181 | 0.274 | 0.180 | 0.274 | 0.180 | 0.272 | 0.182 | 0.274 |
| | 720 | 0.203 | 0.292 | 0.212 | 0.297 | 0.212 | 0.300 | 0.212 | 0.300 | 0.207 | 0.296 | 0.212 | 0.300 | 0.214 | 0.304 |
| | Avg | 0.170 | 0.260 | 0.176 | 0.264 | 0.178 | 0.269 | 0.178 | 0.269 | 0.175 | 0.266 | 0.177 | 0.267 | 0.179 | 0.270 |
| Weather | 96 | 0.163 | 0.202 | 0.170 | 0.208 | 0.206 | 0.257 | 0.208 | 0.259 | 0.207 | 0.252 | 0.209 | 0.258 | 0.171 | 0.210 |
| | 192 | 0.214 | 0.248 | 0.219 | 0.252 | 0.264 | 0.300 | 0.252 | 0.285 | 0.264 | 0.303 | 0.258 | 0.291 | 0.246 | 0.278 |
| | 336 | 0.274 | 0.294 | 0.279 | 0.296 | 0.309 | 0.326 | 0.311 | 0.328 | 0.314 | 0.333 | 0.312 | 0.331 | 0.296 | 0.313 |
| | 720 | 0.351 | 0.344 | 0.358 | 0.347 | 0.377 | 0.369 | 0.374 | 0.364 | 0.384 | 0.377 | 0.383 | 0.373 | 0.362 | 0.353 |
| | Avg | 0.251 | 0.272 | 0.257 | 0.276 | 0.289 | 0.313 | 0.286 | 0.309 | 0.292 | 0.316 | 0.291 | 0.313 | 0.269 | 0.289 |

Table 11: Varying input sequence length results on the Weather dataset.

| Models | | | **Time-o1** | | iTransformer | | **Time-o1** | | PatchTST | |
|---|---|---|---|---|---|---|---|---|---|---|
| Metrics | | | MSE | MAE | MSE | MAE | MSE | MAE | MSE | MAE |
| Input sequence length | 96 | 96 | 0.163 | 0.202 | 0.171 | 0.210 | 0.175 | 0.213 | 0.200 | 0.244 |
| | | 192 | 0.214 | 0.248 | 0.246 | 0.278 | 0.224 | 0.257 | 0.229 | 0.263 |
| | | 336 | 0.274 | 0.294 | 0.296 | 0.313 | 0.276 | 0.296 | 0.287 | 0.303 |
| | | 720 | 0.351 | 0.344 | 0.362 | 0.353 | 0.353 | 0.346 | 0.363 | 0.353 |
| | | Avg | 0.250 | 0.272 | 0.269 | 0.289 | 0.257 | 0.278 | 0.270 | 0.291 |
| | 192 | 96 | 0.163 | 0.205 | 0.168 | 0.215 | 0.158 | 0.199 | 0.164 | 0.208 |
| | | 192 | 0.210 | 0.248 | 0.213 | 0.253 | 0.204 | 0.242 | 0.225 | 0.269 |
| | | 336 | 0.259 | 0.287 | 0.265 | 0.294 | 0.257 | 0.286 | 0.287 | 0.308 |
| | | 720 | 0.334 | 0.338 | 0.341 | 0.345 | 0.332 | 0.337 | 0.341 | 0.345 |
| | | Avg | 0.241 | 0.270 | 0.247 | 0.277 | 0.238 | 0.266 | 0.254 | 0.283 |
| | 336 | 96 | 0.157 | 0.203 | 0.162 | 0.213 | 0.150 | 0.196 | 0.156 | 0.206 |
| | | 192 | 0.199 | 0.246 | 0.211 | 0.256 | 0.196 | 0.241 | 0.222 | 0.277 |
| | | 336 | 0.251 | 0.287 | 0.260 | 0.295 | 0.246 | 0.282 | 0.251 | 0.285 |
| | | 720 | 0.324 | 0.338 | 0.332 | 0.341 | 0.320 | 0.333 | 0.327 | 0.338 |
| | | Avg | 0.233 | 0.268 | 0.241 | 0.276 | 0.228 | 0.263 | 0.239 | 0.277 |
| | 720 | 96 | 0.161 | 0.213 | 0.172 | 0.225 | 0.152 | 0.201 | 0.154 | 0.207 |
| | | 192 | 0.205 | 0.250 | 0.220 | 0.268 | 0.198 | 0.248 | 0.205 | 0.254 |
| | | 336 | 0.254 | 0.292 | 0.282 | 0.311 | 0.248 | 0.284 | 0.248 | 0.288 |
| | | 720 | 0.318 | 0.339 | 0.337 | 0.351 | 0.313 | 0.335 | 0.317 | 0.339 |
| | | Avg | 0.235 | 0.274 | 0.253 | 0.289 | 0.228 | 0.267 | 0.231 | 0.272 |

Table 12: Experimental results ($\mathrm{mean}_{\pm\mathrm{std}}$) with varying seeds (2021-2025).

| Dataset | | ECL | | | | Weather | | | |
|---|---|---|---|---|---|---|---|---|---|
| Models | | **Time-o1** | | DF | | **Time-o1** | | DF | |
| Metrics | | MSE | MAE | MSE | MAE | MSE | MAE | MSE | MAE |
| 96 | | $0.145_{\pm0.000}$ | $0.235_{\pm0.000}$ | $0.150_{\pm0.001}$ | $0.242_{\pm0.001}$ | $0.164_{\pm0.001}$ | $0.203_{\pm0.001}$ | $0.190_{\pm0.012}$ | $0.232_{\pm0.014}$ |
| 192 | | $0.160_{\pm0.001}$ | $0.249_{\pm0.001}$ | $0.166_{\pm0.002}$ | $0.257_{\pm0.002}$ | $0.216_{\pm0.002}$ | $0.250_{\pm0.001}$ | $0.240_{\pm0.011}$ | $0.272_{\pm0.010}$ |
| 336 | | $0.174_{\pm0.002}$ | $0.266_{\pm0.002}$ | $0.181_{\pm0.001}$ | $0.273_{\pm0.001}$ | $0.274_{\pm0.001}$ | $0.294_{\pm0.001}$ | $0.293_{\pm0.003}$ | $0.310_{\pm0.003}$ |
| 720 | | $0.205_{\pm0.001}$ | $0.293_{\pm0.001}$ | $0.216_{\pm0.004}$ | $0.303_{\pm0.003}$ | $0.353_{\pm0.002}$ | $0.344_{\pm0.001}$ | $0.361_{\pm0.002}$ | $0.352_{\pm0.001}$ |
| Avg | | $0.171_{\pm0.001}$ | $0.261_{\pm0.001}$ | $0.178_{\pm0.001}$ | $0.269_{\pm0.001}$ | $0.252_{\pm0.001}$ | $0.273_{\pm0.001}$ | $0.271_{\pm0.003}$ | $0.292_{\pm0.003}$ |

