# OpenReview forum: "Time-o1: Time-Series Forecasting Needs Transformed Label Alignment"
_NeurIPS.cc/2025/Conference — NeurIPS 2025 poster_

### Official Review · Reviewer_nczu · 2025-06-21

**Clarity:** 3
**Significance:** 3
**Originality:** 3
**Rating:** 5
**Confidence:** 3

**Summary:**

Existing methods predominantly utilize the temporal mean squared error, which faces two critical challenges: (1) label autocorrelation, which leads to bias from the label sequence likelihood; (2) excessive amount of tasks, which increases with the forecast horizon and complicates optimization. In this work, the authors propose Transform-enhanced Direct Forecast (TransDF), which transforms the label sequence into decorrelated components with discriminated significance. Models are trained to align the most significant components, thereby effectively mitigating label autocorrelation and reducing task amount.

**Questions:**

Please see Weaknesses!

**Ethical Concerns:**

["NO or VERY MINOR ethics concerns only"]

**Final Justification:**

The author provided more detailed experimental results, which addressed my concerns.

**Limitations:**

Please see Weaknesses!

**Quality:**

3

**Strengths And Weaknesses:**

Strengths:
1. The core idea of applying PCA-like transformations to the label space to mitigate autocorrelation bias and reduce task complexity is novel and impactful. The use of SVD to generate decorrelated components ranked by significance provides a theoretically grounded solution. This approach effectively bridges a gap in objective design, which has been underexplored compared to architectural advancements.
2. The paper validates TransDF rigorously across 8+ datasets, 11+ SOTA baselines, and multiple forecasting models (Transformers, MLPs, etc.). Ablation studies clearly dissect the contributions of decorrelation and task reduction, while sensitivity analyses justify design choices. The consistent improvements (e.g., Fredformer + TransDF reduces MSE by 0.016 on ETTh1) demonstrate broad applicability.

Weaknesses:
1. While Theorem 3.1 states the bias of TMSE under autocorrelation, its proof is relegated to Appendix without intuitive explanation in the main text. The connection between SVD-based decorrelation (Lemma 3.2) and bias elimination (§3.2) lacks formal derivation. A corollary linking Lemma 3.2 to Theorem 3.1’s bias term would strengthen theoretical rigor.
2. TransDF introduces SVD on label sequences during training. While §E.1 (Appendix) mentions efficiency, the main text does not quantify its cost relative to forecast horizon T or compare it to FreDF/Fourier transforms. For long horizons, this could be non-trivial. A complexity analysis for SVD) and wall-clock time comparisons would clarify practical trade-offs.

---

> ### Author Rebuttal · Authors · 2025-07-30
>
> We sincerely appreciate the reviewer for the positive evaluation and appreciation of our novelty, theoretical ground, rigorous experiments, and broad applicability. The raised two weaknesses are concrete and actionable. Please kindly check our responses and actions to address the two weaknesses.
>
> ---
>
>
>
> #### [W1] *While Theorem 3.1 states the bias of TMSE under autocorrelation, its proof is relegated to Appendix without **intuitive explanation** in the main text. The connection between SVD-based decorrelation (Lemma 3.2) and bias elimination (§3.2) lacks formal derivation. **A corollary linking Lemma 3.2 to Theorem 3.1’s bias term** would strengthen theoretical rigor.*
>
> **Response.** We sincerely appreciate the rigorous and helpful suggestion. Below we add (1) an intuitive explanation to the proof of Theorem 3.1, and (2) a corollary demonstrating Lemma 3.2 eliminates bias in Theorem 3.1.
>
> - We provide an intuitive explanation of Theorem 3.1 in the main text (after line 93). The content is presented as the bold fonts below.
> > First, TMSE introduces bias due to autocorrelation. In time-series forecasting, observations exhibit strong dependencies on their past values, resulting in step-wise correlation in the label sequence. In contrast, TMSE treats the forecast of each step as an independent task, thereby neglecting these correlations. This misalignment causes TMSE to be biased relative to the true likelihood of the label sequence, as formalized in Theorem 3.1. **Intuitively, assuming the label sequence obeys a Gaussian distribution; the true joint likelihood should incorporate autocorrelation via off-diagonal elements in covariance matrix $\Sigma$. In contrast, TMSE only considers diagonal elements in $\Sigma$,  ignoring off-diagonal covariances, which makes it biased from the true likelihood except in cases where labels are uncorrelated ($\Sigma$ is diagonal).**
>
>
> - We add a corollary linking Lemma 3.2 to Theorem 3.1’s bias term (after line 139).
>
> >**Corollary 3.4** Let $\mathbf{Z} = \mathbf{Y}\mathbf{P}^*$ denote the decorrelated components in Lemma 3.2. Then, applying TMSE to $\mathbf{Z}$ is equivalent to maximizing their joint likelihood without the bias in Theorem 3.1.
>
> >**Proof sketch.** By Lemma 3.2, the components $\mathbf{Z}$ are decorrelated and have a diagonal covariance matrix $\Sigma=I$. According to Theorem 3.1, the bias term of applying TMSE becomes:
> $
> \mathrm{Bias} = \|\mathbf{Z} - \hat{\mathbf{Z}}\|_{I}^{2} - \|\mathbf{Z} - \hat{\mathbf{Z}}\|^{2} -\frac{1}{2}\log|I| = 0.
> $
> Thus, applying TMSE to $\mathbf{Z}$ provides an unbiased estimation of their likelihood. $\square$
>
>
>
> #### [W2] *TransDF introduces SVD on label sequences during training. While §E.1 (Appendix) mentions efficiency, the main text does not quantify its cost relative to forecast horizon T or compare it to FreDF/Fourier transforms. For long horizons, this could be non-trivial. **A complexity analysis for SVD and wall-clock time comparisons would clarify practical trade-offs.***
>
> **Response.** Thank you very much for your actionable and informative suggestion. You raise a valid concern about the practical efficiency of our method, particularly for long forecast horizons. Below, we provide a comprehensive analysis on **both theoretical complexity and empirical wall-clock time** of TransDF. We also include **FreDF** for comparison.
>
>
> Our TransDF framework involves 3 stages, each with its own computational characteristics:
>
> - In the pre-processing stage, TransDF calculates the projection matrix $\mathbf{P}$ via SVD.
>     - **Complexity analysis.** Suppose $\mathbf{Y}\in\mathbb{R}^{\mathrm{m\times T}}$ is the normalized label sequence ($\mathrm{m}$: sample size,  $\mathrm{T}$: forecast horizon); we first calculate the covariance matrix as $\mathbf{S}=\mathbf{Y}^\top \mathbf{Y}\in\mathbb{R}^{\mathrm{T\times T}}$, where the complexity is $\mathcal{O}(\mathrm{T^2m})$. Then, we apply SVD to $\mathbf{S}$ to obtain projection matrix $\mathbf{P}$, where the complexity is the standard complexity of SVD: $\mathcal{O}(\mathrm{T^3})$ [1]. Therefore, the overall complexity is $\mathcal{O}(\mathrm{T^3}+\mathrm{T^2m})$. In contrast, FreDF does not require pre-processing.
>     - **Wall-clock time analysis.** We investigate wall-clock time of the pre-processing stage on the Weather dataset given varying sample size (m) and prediction horizon (T). The results are presented below with two key notes. (1) the running time increases with m and T, as indicated in complexity analysis. (2) **The pre-processing step only executes once per training process, which makes the cost tolerate given largest m and T (less than 18 seconds)**.
>
>
> |  |   T=96         |   T=192         |   T=336         |   T=720          |
> |-----------|---------------|-----------------|-----------------|------------------|
> |  m=7200    | 0.399$_{\pm0.002}$ | 0.700$_{\pm0.002}$ | 1.361$_{\pm0.002}$ | 3.182$_{\pm0.002}$  |
> |  m=14400    | 0.740$_{\pm0.002}$ | 1.697$_{\pm0.003}$ | 3.157$_{\pm0.006}$ | 6.605$_{\pm0.008}$  |
> |  m=21600    | 1.198$_{\pm0.004}$ | 2.735$_{\pm0.003}$ | 4.771$_{\pm0.007}$ | 10.211$_{\pm0.010}$ |
> |  m=28800    | 1.785$_{\pm0.002}$ | 3.693$_{\pm0.004}$ | 6.450$_{\pm0.003}$ | 13.796$_{\pm0.011}$ |
> |  m=36000    | 2.351$_{\pm0.003}$ | 4.632$_{\pm0.008}$ | 8.159$_{\pm0.005}$ | 17.392$_{\pm0.004}$ |
>
> - In the training stage, during each iteration, TransDF calculates the transformed-domain loss using $\mathbf{P}$. The complexity is $\mathcal{O}(\mathrm{BTK})$. In contrast, the complexity of FreDF is $\mathcal{O}(\mathrm{BT log T})$.
>     - **Complexity analysis.** Suppose $\mathbf{P}\in\mathbb{R}^{\mathrm{T \times K}}$ is the projection matrix obtained in the pre-processing stage ($\mathrm{B}$: the batch size, $\mathrm{T}$: prediction horizon). We first generate latent components as $\mathbf{Z}=\mathbf{YP}$, $\hat{\mathbf{Z}}=\hat{\mathbf{Y}}\mathbf{P}$, where the complexity is $\mathcal{O}(\mathrm{BTK})$. We then compute the L1-loss between $\mathbf{Z}$ and $\hat{\mathbf{Z}}$, where the complexity is $\mathcal{O}(\mathrm{BK})$. Therefore, the overall complexity is $\mathcal{O}(\mathrm{BTK})$, and increases to  $\mathcal{O}(\mathrm{BT^2})$ under the full alignment case ($\mathrm{T=K}$).
>     - In contrast, **FreDF** applies FFT to generate latent components, with complexity $\mathcal{O}(\mathrm{BT log T})$,  which is more efficient than TransDF's $\mathcal{O}(\mathrm{BTK})$ given $\mathrm{K}>\log\mathrm{T}$.
>     - **Wall-clock time analysis.** We investigate wall-clock time of this stage on the Weather dataset. Without loss of generality, we set batch size (B) as 4096 and consider the full alignment case $\mathrm{T=K}$. We compare the time for calculating loss of DF, FreDF and TransDF in the table below and summarize two findings. (1) **TransDF's per-batch overhead is marginally higher than FreDF**, consistent to the complexity analysis. (2) **The absolute magnitude of this overhead is negligible  (≤1.17ms per batch).**
>
> | Method | T=96              | T=192              | T=336              | T=720              |
> |----------|-------------------|--------------------|--------------------|--------------------|
> | DF       | 0.057$_{\pm 0.004}$   | 0.052$_{\pm 0.010}$    | 0.060$_{\pm 0.004}$    | 0.078$_{\pm 0.018}$    |
> | FreDF    | 0.172$_{\pm 0.034}$   | 0.182$_{\pm 0.025}$    | 0.199$_{\pm 0.009}$    | 0.707$_{\pm 0.421}$    |
> | TransDF  | 0.198$_{\pm 0.034}$   | 0.204$_{\pm 0.029}$    | 0.220$_{\pm 0.003}$    | 1.170$_{\pm 0.807}$    |
>
> - **In the inference stage, both FreDF and TransDF have the same complexity with DF.** They operate identically to DF, directly using the forecast model's output sequence with no extra computations.
> - In conclusion, the complexity of TransDF is ignorable in both training and inference stages. For the pre-processing stage that only executes once, the complexity is also limited. It ensures that TransDF remains scalable for long label sequence and large datasets.
>
>
>
>
> **Reference**
>
>
> [1] Tutorial: Complexity analysis of Singular Value Decomposition and its variants. arXiv preprint arXiv:1906.12085, 2019.

---

> > ### Comment · Reviewer_nczu · 2025-08-04
> > **Response**
> >
> > The author addressed my concerns. I believe that the more abundant experimental results further confirm the effectiveness of the proposed method. Therefore, I have decided to increase the score to 5.

---

> > > ### Author Response · Authors · 2025-08-04
> > >
> > > Dear nczu,
> > >
> > > Thank you very much for your prompt response and for your active participation of our work. We sincerely appreciate you adjusting the evaluation score to 5.
> > >
> > > Your support and valuable feedback are highly encouraging to us.
> > >
> > > TransDF Author Team

---

### Official Review · Reviewer_Vg9Y · 2025-07-01

**Clarity:** 4
**Significance:** 3
**Originality:** 3
**Rating:** 5
**Confidence:** 3

**Summary:**

This paper proposes a new label transformation method that transforms time series forecasting labels to decorrelate labels and mitigate the unimportant forecasting labels, so as to reduce forecasting tasks. The proposed method can then enhance the performance of forecasting models and maintain high efficiency. Experiment results show that the proposed method can help improve the performance of existing forecasting models as a plug-and-play module. Comprehensive results further prove the effectiveness of the label decorrelation and reduction of the number of tasks.

**Questions:**

How to interpret the sharp error increase at $\alpha=1$ for some datasets like weather, while other datasets do not?

How does the proposed model perform when applied to non-deep learning forecasting models?

**Ethical Concerns:**

["NO or VERY MINOR ethics concerns only"]

**Final Justification:**

This paper is well written with convincing motivations, clear structures, comprehensive details, as well as a well-structured code repository. The proposed method also presents novelty and effectiveness for the documented existing problems. My main concern about the sharp error increase has been well addressed with detailed explanations and experiments during the rebuttal. I thus advocate for the acceptance of this paper.

**Limitations:**

The paper discusses future work, but does not really discuss limitations, even though it has a "limitations and future works" section.

**Paper Formatting Concerns:**

No formatting concerns

**Quality:**

3

**Strengths And Weaknesses:**

Strengths:
1. The paper is well written with comprehensive details and results.
2. The idea is well motivated and explained. The motivation examples are easy to follow and make sense.
3. The proposed method is efficient but effective. Experiments are designed carefully to show the superiority of the proposed method and to research various aspects of the proposed method.
4. The provided code is comprehensive and structured clearly for researchers to follow.

Weaknesses:
1. Some conclusions are very precise. For example, the paper claims, "We observe that increasing $\alpha$ from 0 to 1 generally leads to improved forecasting accuracy, with the best results typically achieved when $\alpha$ is close to 1." However, from the results, the best results occur at $\alpha=1$ for ETTh2, ETTm1, and ETTm2. This should be presented precisely and explained clearly in the paper.
2. The parameter sensitivity on different tasks shows different patterns, with some datasets showing a decrease with $\alpha$ increases, and on other datasets showing a first decrease and sharp increase at $\alpha=1$. These inconsistent patterns indicate potential failure of unexpected behaviours in some cases.
3. To demonstrate the effectiveness of the method as a plug-and-play module, it would be better to see whether the proposed method improves forecasting performance on more base forecasting models, like non-deep learning ones, instead of four pure deep learning forecasting models.
4. The paper does not really discuss limitations, although it has a section called "limitations and future works".

---

> ### Author Rebuttal · Authors · 2025-07-31
>
> Thank you so much for the positive comments and appreciation of our writing, comprehensive experiments, meaningful motivation, effective methodology and structured code repository.
> The four points raised are very insightful.
> Please kindly check our responses to further address associated concerns.
>
> We observed that some comments in the weaknesses and questions sections convey similar points. We will selectively merge them to make the response concise and easy to follow.
>
> ---
>
>
> #### *[W1] Some conclusions are very precise. For example, the paper claims, "We observe that increasing from 0 to 1 generally leads to improved forecasting accuracy, with the best results typically achieved when $\alpha$ is close to 1." However, from the results, the best results occur at for ETTh2, ETTm1, and ETTm2. This should be presented precisely and explained clearly in the paper.*
> **Response.** Thank you very much for your meticulous feedback! We appreciate your careful review of our experimental results and analysis. In response, we have revised the claim as follows:
> >We observe that increasing α from 0 to 1 generally leads to improved forecasting accuracy. The best results are often achieved as α approaches 1, and in some cases, specifically when  α=1 (e.g., ETTh2, ETTm1, ETTm2).`
>
>
> #### *[W2, Q1] The parameter sensitivity on different tasks shows different patterns, with some datasets showing a decrease with increases, and on other datasets showing a first decrease and sharp increase at $\alpha=1$. These inconsistent patterns indicate potential failure of unexpected behaviors in some cases. How to interpret the sharp error increase at $\alpha=1$ for some datasets like weather, while other datasets do not?*
>
> **Response.** This phenomenon also raises our attention. We checked the phenomenon and identify the issues via bad case analysis.
> - We conducted a case study of the extracted components on Weather and other datasets with normal behavior. **The component magnitudes in the Weather dataset are significantly lower than in other datasets.** For example, fixing the forecast horizon T=720, mmost components on the Weather dataset fluctuate within [-2,2], with informative non-top components close to zero. In contrast, components on the ETTh1 dataset typically range from -40 to 10 and show larger values of non-top components. **This small amplitude is unique to datasets that exhibit a sharp error increase at $\alpha=1$ (such as Weather).**
> - Therefore, we hypothesize that this sharp increase at $\alpha=1$ is **caused by the small amplitude of components**, resulting in diminished losses and gradients, which hinders the effective alignment of important yet low-magnitude components.
> - One strategy to counteract the small amplitude of components is increasing learning rate. We increase the learning rate to 0.005 and investigate the sensitivity of $\alpha$ on Weather dataset once again. The results are presented below, where the **sharp error increase at $\alpha=1$ is effectively alleviated.**
>
> |              | T=96   | T=96   | T=192  | T=192  | T=336  | T=336  | T=720  | T=720  | Avg    | Avg    |
> | ------------ | ------ | ------ | ------ | ------ | ------ | ------ | ------ | ------ | ------ | ------ |
> | Metric       | MAE    | MSE    | MAE    | MSE    | MAE    | MSE    | MAE    | MSE    | MAE    | MSE    |
> | $\alpha=0$   | 0.228  | 0.174  | 0.266  | 0.213  | 0.316  | 0.270  | 0.362  | 0.337  | 0.293  | 0.249  |
> | $\alpha=0.2$ | 0.226  | 0.172  | 0.262  | 0.212  | 0.304  | 0.263  | 0.361  | 0.335  | 0.288  | 0.245  |
> | $\alpha=0.4$ | 0.225  | 0.172  | 0.266  | 0.213  | 0.307  | 0.264  | 0.353  | 0.331  | 0.288  | 0.245  |
> | $\alpha=0.6$ | 0.223  | 0.172  | 0.265  | 0.213  | 0.308  | 0.264  | 0.355  | 0.332  | 0.288  | 0.245  |
> | $\alpha=0.8$ | 0.220  | 0.172  | 0.259  | 0.211  | 0.301  | 0.261  | 0.362  | 0.332  | 0.285  | 0.244  |
> | $\alpha=1.0$ | 0.216  | 0.169  | 0.258  | 0.213  | 0.302  | 0.264  | 0.351  | 0.331  | 0.282  | 0.244  |
>
> #### *[W3, Q2] To demonstrate the effectiveness of the method as a plug-and-play module, it would be better to see whether the proposed method improves forecasting performance on more base forecasting models, like **non-deep learning ones**, instead of pure deep learning forecasting models. How does the proposed model perform when applied to non-deep learning forecasting models?*
>
> **Response.** Thank you for your insightful comment and suggestion. We agree that clarifying the applicability of TransDF to non-deep learning methods is important. However, the challenges that TransDF addresses—label correlation bias and excessive task amount—are generally absent in most non-deep learning approaches, as outlined below:
> - **Firstly, TransDF is designed to improve the multitask, multi-step forecasting paradigm**, where multiple future values are predicted simultaneously. **The two challenges inherently arise from this multitask setting** (label correlation bias and excessive task amount).
> - **In contrast, non-deep learning methods with defined learning objectives are often implemented for single-step forecasting**. Examples include linear regression (LR) and gradient boosted decision trees (GBDT)~[1]. Although these methods can be extended to multi-step forecasting (e.g., via iterative prediction or by training T separate models for T steps), the resulting formulations remain single-task, with each model trained independently for one step ahead.
> -  **In single-task learning, the aforementioned challenges do not emerge**. Since only one label is predicted at a time, neither label correlation nor the need to reduce the number of tasks is relevant. Consequently, **applying TransDF to such settings is unnecessary.**
> - We agree it is important to clarify that TransDF and its targeted challenges are specific to multitask forecasting methods that are often deep models. **We have incorporated it to make our limitation section more concrete.** Please see the response to [W4] for details.
>
>
> #### *[W4] The paper does not really discuss limitations, although it has a section called "limitations and future works".*
>
> **Response.** Once again, we sincerely appreciate the kind and actionable feedback provided. In response, **we have expanded the “Limitations and Future Work” section** by incorporating the points discussed in [W3, Q2], making our discussion more concrete. The revised text is as follows:
> > Limitations & future works. TransDF aims to address label correlation bias and excessive task amount —challenges that are relevant to multi-step forecasting. In single-step forecasting scenarios such as streaming prediction, where these challenges are less relevant, TransDF might be not applicable. Additionally, this study assumes that latent components with higher variance are more significant. While generally effective, this assumption may not hold in scenarios where crucial information is contained in low-variance component. For example, in anomaly forecasting, subtle patterns indicative of anomalies may exhibit small variance. A promising direction for future work is to develop an adaptive projection strategy, possibly through meta-learning, that directly targets downstream task performance.

---

> > ### Comment · Reviewer_Vg9Y · 2025-08-01
> >
> > I appreciate the authors' efforts for the rebuttal. The responses have addressed my concerns, especially the sharp error increase issue. I would maintain a positive attitude for this paper.

---

> ### Author Response · Authors · 2025-08-01
>
> Dear Vg9Y,
>
> We are pleased that our responses address your concerns!
>
> Thank you very much for reviewing our response and providing continued support.
>
> TransDF Author Team

---

### Official Review · Reviewer_2azR · 2025-07-03

**Clarity:** 3
**Significance:** 3
**Originality:** 2
**Rating:** 4
**Confidence:** 3

**Summary:**

TransDF proposes a novel training objective that decorrelates the label space and aligns the most informative components to mitigate label autocorrelation and reduce the task amount. It can be plugged into existing time-series forecasting (TSF) models by reformulating the loss objective and applying an SVD-based label transformation.

**Questions:**

Why does even TransDF† (which uses a random projection and only partial component alignment) outperform the canonical DF objective, especially on the Weather dataset?

**Ethical Concerns:**

["NO or VERY MINOR ethics concerns only"]

**Final Justification:**

Overall, I appreciate the author's detailed responses and additional experiments, which have clarified my main concerns on local projection strategies and partial component alignment. My raised issues are largely addressed, and I find my original assessment remains appropriate, so I maintain it as it is. Although I could not engage more actively in the discussion phase, I found the author's rebuttal convincing and believe my initial review and score remain appropriate.

**Limitations:**

The SVD-based label transformation relies solely on label-side prior statistics, making it less suitable for fully online or streaming scenarios. This limitation may hinder deployment in real-time TSF systems, where adaptive behavior and incremental updates are often required.

**Quality:**

3

**Strengths And Weaknesses:**

Strengths
- The paper clearly formulates the bias issue of the TMSE objective in Theorem 3.1 and addresses it by aligning fully decorrelated label components through SVD-based transformation.
- While FreDF relies on the assumption of decorrelation between frequency bases,which only holds under infinite forecast horizons, TransDF removes label autocorrelation bias under realistic finite horizons and simultaneously reduces the number of forecasting tasks.

Weaknesses
- The proposed method applies a global projection matrix derived from SVD uniformly across all forecast steps. However, real-world time-series data often contain local pattern shifts or nonlinear/dynamic structures. Thus, a static global projection may fail to capture such local dynamics, which might explain smaller improvements on certain datasets (e.g., PEMS or Traffic).

---

> ### Author Rebuttal · Authors · 2025-07-29
>
> Thank you very much for your positive comments and appreciation of our contributions. Below are our responses to the specific query raised.
>
> ---
>
> #### *[W1] The proposed method applies a global projection matrix derived from SVD uniformly across all forecast steps. However, real-world time-series data often contain local pattern shifts or nonlinear/dynamic structures. Thus, a static global projection may fail to capture such local dynamics, which might explain smaller improvements on certain datasets (e.g., PEMS or Traffic).*
>
> **Response**. Thank you for this kind and informative comments! We fully agree that `a static global projection may fail to capture such local dynamics`, which is an important direction for future research. Nevertheless, **we slightly note that the static global projection suffices to address the two research problems in the current work.** Please look at the concise analysis below.
> - **In this work, we target two challenges of designing learning objectives for time-series forecasting models:** (1) label correlation, which leads to bias from the true label likelihood; (2) excessive amount of tasks, which increases with the forecast horizon and complicates optimization.
> - **The global projection approach in Eq. (4) suffices to address the two challenges theoretically (Lemma 3.2) and empirically (Table 3)—our primary focus in this work.**
> - We concur that developing adaptive or local projection strategies can enhance modeling of dynamic and nonlinear structures. Nevertheless, it remains non-trivial to construct such methods while preserving the rigor of bias elimination and effective task reduction, **suggesting a promising avenue** for future exploration. One promising approach, for instance, may consider learnable projections with appropriate regularization to ensure decorrelation for debiasing and significance discrimination for reducing task amount.
>
> #### *[Q1] Why does even TransDF† (which uses a random projection and only partial component alignment) outperform the canonical DF objective, especially on the Weather dataset?*
> **Response**. This scenario also raises our attention. We attribute the superior performance of TransDF† over the canonical DF objective primarily to the reduction in optimization tasks.
>
> - **Firstly, partial alignment (aligning only a subset of components) introduces two opposing effects:** (1) some information is lost by discarding components, which may hinder learning and **reduce performance**, and (2) the number of tasks to optimize is reduced, which can reduce optimization difficulty and **improve performance**. The latter reflects a common issue in multitask learning: an excessive number of tasks often causes **gradient conflict** and hampers training [1,2].
> - **Secondly, the opposing impacts above make partial component alignment plausible to improve performance.** On representative datasets like Weather, tuning the number of aligned components allows partial alignment to outperform full alignment, suggesting that the benefits of reduced task conflict outweigh the drawbacks of partial information loss in this setting.
>
>
> #### *[L1] The SVD-based label transformation relies solely on label-side prior statistics, making it less suitable for fully online or streaming scenarios. This limitation may hinder deployment in real-time TSF systems, where adaptive behavior and incremental updates are often required.*
>
> **Response.** Thank you very much for highlighting this important aspect.
> - In this study, we investigate **two central challenges in multi-step (multi-task) forecasting**: (1) label correlation bias and (2) excessive task amount. We develop TransDF to address the two challenges.
> - We acknowledge that **in the training stage**, TransDF requires access to the full T-step ahead label sequence for the SVD-based label transformation. **This requirement makes TransDF less suitable for training scenarios where only 1-step ahead labels are available, such as fully online or streaming forecasting.**
>   - By the way, if complete T-step ahead label sequences are available during training, TransDF remains applicable. Importantly, during inference, the SVD transformation is not needed, so TransDF can be deployed on streaming data; for 1-step-ahead online forecasting, the first prediction step can be directly utilized.
> - Nevertheless, in standard online or streaming tasks, **the two challenges addressed in this work do not arise**, as these challenges are specific to multi-step forecasting scenarios with label sequences longer than one. Therefore, employing TransDF in fully online or streaming settings is unnecessary.
> - We agree it is important to clarify that TransDF and its targeted challenges are specific to multi-task (multi-step) forecasting scenarios. The enhanced limitation section is presented as follows.
> > Limitations & future works. **TransDF aims to address label correlation bias and excessive task amount —challenges that are relevant to multi-step forecasting. In single-step forecasting scenarios such as streaming prediction, where these challenges are less relevant, TransDF might be not applicable.** Additionally, this study assumes that latent components with higher variance are more significant. While generally effective, this assumption may not hold in scenarios where crucial information is contained in low-variance component. For example, in anomaly forecasting, subtle patterns indicative of anomalies may exhibit small variance. A promising direction for future work is to develop an adaptive projection strategy, possibly through meta-learning, that directly targets downstream task performance.

---

> > ### Comment · Area_Chair_3iAz · 2025-08-05
> > **Engaging with the rebuttal**
> >
> > Dear reviewer,
> >
> > The discussion phase is soon coming to and end. It will be great if you could go over the rebuttal and discuss with the authors if you still have outstanding concerns. Thank you for being part of the review process.
> >
> > Regards,
> >
> > Area Chair

---

> ### Author Response · Authors · 2025-08-05
>
> Dear 2azR,
>
> Once again, we express our sincere gratitude on the constructive feedback provided in the weaknesses and questions sections. In this window, we would like to offer supplementary responses and results to further clarify the two primary concerns (W1 and Q1).
>
> #### **[W1] Real-world time-series data often contain local pattern shifts or nonlinear/dynamic structures. Thus, a static global projection may fail to capture such local dynamics, which might explain smaller improvements on certain datasets (e.g., PEMS or Traffic).**
> **Response.** We would like to further clarify this aspect with an intuitive and evidence-based discussion as follows:
> - Firstly, this paper identifies two challenges of designing learning objectives for time-series forecasting models. **The global projection method is devised to address the two challenges, offering theoretical and empirical supports** to demonstrate these claims.
> - Secondly, we agree that local projection strategies hold huge potentials. We add experiments to include incorporating two well-known local methods: the Short-Term Fourier Transform (STFT) and the Discrete Wavelet Transform (DWT). We replaced our global projection with these local strategies and compared their performance against both the baseline (no projection) and our proposed TransDF. The results are in the table below, leading to two key observations:
>     - **STFT and DWT projections are effective.** Your intuition was correct—both STFT and DWT improve forecasting performance over the baseline (DF). This confirms that capturing local or dynamic patterns is valuable and provides a performance improvement. We are grateful for this suggestion, as it adds a new dimension to our analysis.
>     - **TransDF outperforms STFT and DWT.** Despite the benefits of STFT and DWT, TransDF still outperforms them. This suggests that while capturing local patterns is helpful, the two core challenges we identified (label correlation bias and excessive task amount) have a more significant impact on performance. The success of our global method highlights the importance of addressing these specific issues, which the local strategies do not resolve.
>
>
> | Dataset | Horizon | DF | | STFT | | DWT | | TransDF | |
> | :--- | :--- | :--- | :--- | :--- | :--- | :--- | :--- | :--- | :--- |
> | | | **MSE** | **MAE** | **MSE** | **MAE** | **MSE** | **MAE** | **MSE** | **MAE** |
> | **ETTh1** | 96 | 0.377 | 0.396 | *0.371* | 0.392 | 0.371 | *0.391* | **0.368** | **0.391** |
> | | 192 | 0.437 | 0.425 | *0.429* | *0.423* | 0.430 | 0.424 | **0.424** | **0.422** |
> | | 336 | 0.486 | 0.449 | *0.477* | *0.443* | 0.477 | 0.450 | **0.467** | **0.441** |
> | | 720 | 0.488 | 0.467 | *0.476* | *0.466* | 0.477 | 0.469 | **0.465** | **0.463** |
> | | **Avg** | 0.447 | 0.434 | *0.438* | *0.431* | 0.439 | 0.434 | **0.431** | **0.429** |
> | **ECL** | 96 | 0.150 | 0.242 | *0.147* | 0.238 | 0.147 | *0.237* | **0.145** | **0.235** |
> | | 192 | 0.168 | 0.259 | *0.163* | 0.254 | 0.163 | *0.251* | **0.159** | **0.249** |
> | | 336 | 0.182 | 0.274 | *0.176* | 0.268 | 0.176 | *0.266* | **0.173** | **0.264** |
> | | 720 | 0.214 | 0.304 | 0.210 | 0.297 | *0.206* | *0.294* | **0.203** | **0.292** |
> | | **Avg** | 0.179 | 0.270 | 0.174 | 0.264 | *0.173* | *0.262* | **0.170** | **0.260** |
> | **Weather**| 96 | 0.174 | 0.228 | *0.172* | 0.223 | 0.173 | *0.220* | **0.169** | **0.219** |
> | | 192 | 0.213 | 0.266 | *0.212* | 0.264 | 0.212 | *0.259* | **0.210** | **0.258** |
> | | 336 | 0.270 | 0.316 | *0.262* | *0.300* | 0.264 | 0.301 | **0.259** | **0.297** |
> | | 720 | 0.337 | 0.362 | *0.330* | 0.353 | 0.332 | *0.352* | **0.327** | **0.349** |
> | | **Avg** | 0.249 | 0.293 | *0.244* | 0.285 | 0.245 | *0.283* | **0.241** | **0.280** |

---

> ### Author Response · Authors · 2025-08-05
>
> #### **[Q1] Why does even TransDF† (which uses a random projection and only partial component alignment) outperform the canonical DF objective, especially on the Weather dataset?**
>
> **Response.** This scenario also raises our attention. We would like to address this concern with additional experiments and result-driven discussion as follows.
>
> - Firstly, we involve three implementations differing in projection matrices and output dimensions. DF sets the projection matrix $\mathbf{P}$ as identity matrix (i.e., using the temporal MSE loss). TransDF$^*$ sets $\mathbf{P}$ as random matrix with the shape $\mathbb{R}^{\mathrm{T}\times\mathrm{T}}$. TransDF$^\dagger$ further sets the shape as $\mathbb{R}^{\mathrm{T}\times\mathrm{K}}$, i.e., considering $\mathrm{K}$ components for partial alignment. The comparison results are attached in the table below.
> - Secondly, we analyze the results and discern the role of random projection and partial alignment.
>     - **Random projection provides minimal improvement.** Empirically, comparing TransDF* and DF, the performance is comparable. That is, simply replacing the identity matrix with a full random projection does not yield significant improvements.
>     - **Partial alignment outperforms full alignment in some cases.** Empirically, comparing TransDF* and TransDF†, TransDF† outperforms TransDF* in some datasets, despite the improvement is relatively small. Theoretically, the improvement is attributed to the reduction of task amount. By aligning fewer components, the model may experience fewer conflicting gradient signals, leading to better overall forecasting performance. However, this improvement is often modest because the random nature of the projection can also lead to some information loss.
>
> | Model | Projection | Alignment | Data | Avg | |
> | :--- | :---: | :---: | :--- | :--- | :--- |
> | | | | | **MSE** | **MAE** |
> | DF | Identity | Full |  ETTh1 | 0.447 | **0.434** |
> | | | | ETTh2 | 0.377 | 0.402 |
> | | | | ECL | 0.179 | 0.270 |
> | | | | Traffic | 0.426 | 0.285 |
> | | | | Weather | 0.249 | 0.293 |
> | TransDF$^*$ | Random | Full |  ETTh1 | 0.440 | 0.435 |
> | | | | ETTh2 | 0.372 | 0.393 |
> | | | | ECL | 0.177 | 0.265 |
> | | | | Traffic | 0.427 | 0.281 |
> | | | | Weather | 0.248 | 0.283 |
> | TransDF$^\dagger$ | Random | Partial |  ETTh1 | **0.440** | 0.436 |
> | | | | ETTh2 | **0.368** | **0.392** |
> | | | | ECL | **0.175** | **0.264** |
> | | | | Traffic | **0.426** | **0.281** |
> | | | | Weather | **0.244** | **0.281** |
>
> ---
> If our response and additional experiments have resolved the raised issues, we politely invite you to further revise the score.
>
> I personally really enjoy the discussion with you. I am particularly grateful for your thoughtful feedback on local projection strategies, which exhibits promising potential through our preliminary experiments.
>
> If you have any further question and concern, we are happy to discuss.
>
> BG, TransDF Author Team

---

### Official Review · Reviewer_G79D · 2025-07-03

**Clarity:** 3
**Significance:** 2
**Originality:** 2
**Rating:** 5
**Confidence:** 4

**Summary:**

The paper tackles a fundamental but underexplored aspect of neural time‑series forecasting, i.e., the choice of learning objective. Building on the standard direct‑forecast paradigm (which minimizes pointwise temporal MSE), the authors identify two critical shortcomings of this objective in the forecasting setting, Label autocorrelation bias and Excessive multitask complexity. To address these, the authors propose TransDF, which can be dropped in to any existing architecture without structural changes. Extensive experiments across multiple benchmark datasets (ETT, ECL, Weather, M4) and forecasting models (Transformers, MLPs, GNNs) demonstrate that TransDF outperforms both the canonical TMSE loss and a variety of shape‑alignment or frequency‑domain objectives, achieving new state‑of‑the‑art results. An ablation study further isolates the benefits of decorrelation versus task reduction, confirming that their combination is key to TransDF’s success .

**Questions:**

See Weaknesses.

**Ethical Concerns:**

["NO or VERY MINOR ethics concerns only"]

**Final Justification:**

The rebuttal has well addressed my concerns.

**Limitations:**

The Limitations section is concise but could better discuss potential failure modes, e.g., cases where label autocorrelation structure is nonstationary or when small-variance components carry critical signals (anomaly forecasting).

**Paper Formatting Concerns:**

N/A.

**Quality:**

3

**Strengths And Weaknesses:**

## Strengths:
1. The proposed transformation is grounded in sound linear‑algebraic principles, which directly targets the source of bias and task redundancy.
2. The method is model‑agnostic and “drop‑in”—it requires no architectural changes and gracefully extends to any direct‑forecasting network, as evidenced by consistent improvements on FredFormer, iTransformer, FreTS, DLinear, etc.
3. TransDF’s randomized, data‑adapted projections guarantee full decorrelation at finite horizons and discriminate component importance, which is novel to me.

## Weaknesses:
1. The paper could more deeply explore or theoretically characterize when and why certain projection strategies might outperform the randomized approach.
2. The Limitations section is concise but could better discuss potential failure modes, e.g., cases where label autocorrelation structure is nonstationary or when small-variance components carry critical signals (anomaly forecasting).
3. The core idea of transforming labels into another basis has precedents in frequency‑domain losses (FreDF) and shape‑alignment objectives; the manuscript could more explicitly delineate how TransDF’s theoretical guarantees and practical gains substantially surpass these earlier approaches beyond empirical metrics.

---

> ### Author Rebuttal · Authors · 2025-07-29
>
> Thank you very much for your encouraging support and appreciation of our significance, implementation, and novelty. Below are our responses to the specific query raised.
>
> ---
>
> #### *[W1] The paper could more **deeply explore or theoretically characterize** when and why certain projection strategies might outperform the randomized approach.*
>
> **Response.** Thank you for this constructive suggestion. We have now included both **theoretical characterization and empirical exploration** to clarify when and why certain projection strategies outperform randomized approaches.
> Specifically, as shown in the table below, we compare five variants differing in projection matrices $\mathbf{P}$ and output dimensions (full $\mathrm{T}$ versus reduced $\mathrm{K} < \mathrm{T}$).
>
> |Index| Model  | Projection Type | Projection Dimension | ECL | | | | Weather | | | | ETTh1 | | | |
> |-|--|-|-|--|--|--|-|--|--|-|--|--|-|--|--|
> |  | | | | MSE  | $\Delta$ | MAE | $\Delta$ | MSE | $\Delta$ | MAE | $\Delta$ | MSE | $\Delta$ | MAE | $\Delta$ |
> |1| DF | Identity | T | 0.179 | -  | 0.270 | -  |0.249 | -  | 0.293 | -  | 0.447 | - | 0.434 | - |
> |2| TransDF$^{\star}$ | Random | T | 0.177 | 1.1%$\downarrow$ | 0.265 | 1.9%$\downarrow$ | 0.248 | 0.4%$\downarrow$ | 0.283 | 3.4%$\downarrow$ | 0.440 | 1.5%$\downarrow$ | 0.435 | 0.2%$\uparrow$ |
> |3| TransDF$^\dagger$  | Random | K | 0.175 | 2.2%$\downarrow$ | 0.264 | 2.2%$\downarrow$ | 0.244 | 2.0%$\downarrow$ | 0.281 | 4.1%$\downarrow$ | 0.440 | 1.5%$\downarrow$ | 0.435 | 0.2%$\uparrow$ |
> |4| TransDF$^\ddagger$  | Orthogonal | T | 0.172 | 3.9%$\downarrow$ | 0.263 | 2.5%$\downarrow$ | 0.244 | 2.0%$\downarrow$ | 0.283 | 3.4%$\downarrow$ | 0.439 | 1.7%$\downarrow$ | 0.431 | 0.7%$\downarrow$ |
> |5| TransDF  | Orthogonal | K | **0.170** | 5.0%$\downarrow$ | **0.260** | 3.8%$\downarrow$ | **0.241** | 3.2%$\downarrow$ | **0.280** | 4.4%$\downarrow$ | **0.431** | 3.5%$\downarrow$ | **0.429** | 1.1 %$\downarrow$|
>
> - Firstly, we recall the implementation of the 5 projection strategies in the table above. We perform projection on the label sequence $\mathbf{Y}\in\mathbb{R}^{m\times\mathrm{T}}$ and prediction sequence $\hat{\mathbf{Y}}\in\mathbb{R}^{m\times\mathrm{T}}$ to obtain components: $\mathbf{Z}=\mathbf{YP}$,
> $\hat{\mathbf{Z}}=\hat{\mathbf{Y}}\mathbf{P}$, where $\mathbf{P}\in\mathbb{R}^{\mathrm{T}\times\mathrm{K}}$, m is the number of samples, $\mathrm{T}$ is prediction horizon, $\mathrm{K}$ is the number of extracted components.
>     - DF sets $\mathbf{P}$ as identity matrix (i.e., using the temporal MSE loss)
>     - TransDF$^{\star}$ sets $\mathbf{P}$ as random matrix with the shape $\mathbb{R}^{\mathrm{T}\times\mathrm{T}}$. TransDF$^\dagger$ further sets the shape as $\mathbb{R}^{\mathrm{T}\times\mathrm{K}}$, i.e., considering $\mathrm{K}$ components.
>     - TransDF$^\ddagger$ sets $\mathbf{P}$ as the orthogonal matrix calculated via SVD in Lemma 3.3, with the shape $\mathbb{R}^{\mathrm{T}\times\mathrm{T}}$. TransDF further sets the shape as $\mathbb{R}^{\mathrm{T}\times\mathrm{K}}$, i.e., considering $\mathrm{K}$ components.
> - Secondly, we discuss when and why certain projection strategies might outperform the randomized approach (TransDF$^{\star}$).
>     - **Orthogonal projection outperforms random projection.** **Empirically**, comparing line 2 and 4, TransDF$^{\ddagger}$ outperforms TransDF$^\star$; comparing line 3 and 5, TransDF outperforms TransDF$^\dagger$. **Theoretically**, the improvement is attributed to the bias elimination effect of the orthogonal projection matrix (see Lemma 3.2), which enhances learning objective estimation accuracy and therefore improving forecast performance.
>     - **Reduced-dimension projection outperforms full-dimension projection.** **Empirically**, comparing line 2 and 3, TransDF$^{\dagger}$ outperforms TransDF$^\star$; comparing line 4 and 5, TransDF outperforms TransDF$^\ddagger$.  comparing line 2 and 3, 4 and 5. **Theoretically**, the improvement is attributed to the reduced task amount, which mitigates gradient conflicts across different tasks and therefore improving forecast performance.
>
> #### *[W2] The Limitations section is concise but could better discuss potential failure modes, e.g., cases where label autocorrelation structure is nonstationary or when small-variance components carry critical signals (anomaly forecasting).*
>
> **Response.** We agree that the limitations section should be more detailed. The two points raised are insightful. We discuss them below and present a revised limitations section accordingly.
> - **On the case where label autocorrelation structure is nonstationary.** We observe that TransDF retains effective to reduce bias given nonstationary label autocorrelation structure. **Theoretically**, the derivation of decorrelation does not rely on the stationary autocorrelation structure (see Lemma 3.2).
> **Empirically**, for general time-series data that are not always stationary, the label autocorrelation can be effectively eliminated by the proposed method, with the correlation coefficient matrix being a diagonal matrix (see Figure 4 in appendix). **Therefore, TransDF remains applicable when label autocorrelation structure is non-stationary**, since it can theoretically and empirically reduce the label correlation and therefore the bias.
> - **On the case where small-variance component matters.** Our method assumes that components with larger variance are more significant, and thus prioritizes these in the projection. However, we acknowledge that this heuristic may be suboptimal where critical information is embedded in low-variance components. For example, in anomaly forecasting, the small-variance components associated with anomalies may carry low-variance but critical patterns. **We truly appreciate this point and incorporate it to strengthen `limitation section`.**
> > Limitations & future works.  TransDF aims to address label correlation bias and excessive task amount —challenges that are relevant to multi-step forecasting. In single-step forecasting scenarios such as streaming prediction, where these challenges are less relevant, TransDF might be not applicable. **Additionally, this study assumes that latent components with higher variance are more significant. While generally effective, this assumption may not hold in scenarios where crucial information is contained in low-variance component.** For example, in anomaly forecasting, subtle patterns indicative of anomalies may exhibit small variance. A promising direction for future work is to develop an adaptive projection strategy, possibly through meta-learning, that directly targets downstream task performance.
>
>
>
> #### *[W3] The core idea of transforming labels into another basis has precedents in frequency‑domain losses (FreDF) and shape‑alignment objectives; the manuscript could more explicitly delineate how TransDF’s **theoretical guarantees and practical gains** substantially surpass these earlier approaches beyond empirical metrics.*
>
> **Response.** Thank you for your prompting reminder on related works. We agree that it is necessary to further clarify the advantage of TransDF over FreDF and shape-alignment methods. Please find our concise resposne to address this concern below.
> - Firstly, we summarize key advantages of TransDF along two axes in the review report's suggestion.
>   - **Theoretical guarantees.** TransDF eliminates autocorrelation bias in **finite-horizon** settings, while FreDF requires **infinite horizons** and shape-alignment methods provide **no bias elimination guarantees**.
>   - **Practical gains.** TransDF **both** eliminates bias (improving learning objective estimation) and reduces optimization tasks (mitigating optimization difficulties such as gradient conflict), while existing methods address **neither** (shape-alignment) or only **one aspect partially** (FreDF).
>
>
> - Secondly, to provide more comprehensive response, we introduce and compare the three methods considering three factors: (1) whether they formally define the autocorrelation bias, (2) whether they effectively eliminate it, and (3) whether they reduce the amount of optimization tasks. The comparison is summarized below:
>     - **Shape alignment methods** mainly use the DTW loss for intuitive shape matching but **lack autocorrelation bias analysis**. DTW may actually **increase task amount** by matching single predictions to multiple labels.
>     - **FreDF** uses the frequency MAE loss as training objective. This work **identifies the autocorrelation bias under first-order Markov assumptions** but **provides bias elimination guarantees only as $T\rightarrow \infty$** (see Theorem 3.3, FreDF). In real-world settings with finite horizon, FreDF fails to eliminate bias. Moreover, frequency domain transformation preserves the label sequence's length and does not reduce the number of optimization tasks.
>     - **TransDF** **generalizes autocorrelation bias beyond first-order Markov assumptions** and **ensures bias elimination without requiring $T\rightarrow \infty$**, making it practically effective. It also **reduces task amount** by focusing on significant components.
>
> |Method|Autocorrelation bias formalization|Bias elimination|Task reduction|
> |-|-|-|-|
> |Shape alignment|$\times$|$\times$|$\times$|
> |FreDF|$\checkmark$|$\circ$|$\times$|
> |TransDF (Ours)|$\checkmark$|$\checkmark$|$\checkmark$|

---

> > ### Comment · Reviewer_G79D · 2025-08-03
> > **Response to Rebuttal**
> >
> > Authors have properly addressed my concerns, and I decide to raise my score to 5.

---

> ### Author Response · Authors · 2025-08-03
>
> Dear G79D,
>
> Thank you so much for providing swift response and for your dedication in adjusting the evaluation score to 5!
>
> We truly appreciate your increased support.
>
> TransDF Author Team

---

### Decision · Program_Chairs · 2025-09-17

**Decision:**

Accept (poster)

**Comment:**

This work proposes a novel training objective that focuses on the decorrelation of the label space and aligning the most informative components to mitigate label autocorrelation and reduce the task amount. The work tackles a very important problem and was well received by the 4 reviewers. The overall motivation, the model agnostic nature and the core idea of the solution was well appreciated in the initial reviews. There were some major points of contention which included:

* Lack of a theoretical justification of the framework.
* Limitations of the work not adequately addressed.
* SInce the model focuses on obtaining a static global projection this may fail to capture the inherent local dynamics that are unique (to an extent) within different data sets.

The rebuttal provided by the authors was pretty comprehensive and was well appreciated by the reviewers. I read the paper and feel that this is an important contribution and will benefit from being presented in the conference. I recommend acceptance and request the authors to incorporate the rebuttal in the final version of the manuscript.